# Research on adaptive hydraulic drive optimization control of concrete mixing tank truck for open-pit mine

**Guangwei Liu[1], Chonghui Ren[1]\*, Senlin Chai[1], Xuedong Wang[1], Wei Liu[2]**

**1** College of Mining, Liaoning Technical University, Fuxin, Liaoning, China, **2** College of Science, Liaoning Technical University, Fuxin, Liaoning, China

\* renchonghui1229@163.com

## Abstract

The non-axisymmetric problem caused by the fluid sloshing in the tank of a mining concrete mixing tank truck during driving is affected by the excitation of complex road surfaces. The fluid sloshing is coupled with the dynamics of the vehicle body due to the excitation of the complex road surface. The traditional hydraulic drive proportional integral differential (PID) control method is not effective in dealing with such problems, which can easily lead to accidents such as overturning. To improve the accuracy and stability of the hydraulic drive control system, this paper proposes an optimized particle filter PID adaptive control method based on the elastic firefly (FA) algorithm to accelerate the convergence speed of control parameter optimization, and then analyzes its hydraulic drive control characteristics and structural applications, and discusses step steering and double lane change modes are simulated under filling rates of 1.5 and 2.0, respectively. The experimental results show that compared with traditional PID control, the proposed adaptive control method can significantly reduce the average speed error of hydraulic drive control to 0.03km/h and the maximum speed error to 0.17km/h. It also improves the control tracking performance and stability. The practicality of the adaptive hydraulic drive is verified in the filling rate experiments under step steering and double-lane shifting conditions. It has important reference value for the practical application of hydraulic drive control optimization of mining concrete mixing transport tank trucks.

## 1. Introduction

The mining site of an open-pit mine is usually vast, with an open working surface and complex environmental conditions in the mining area. It often faces various harsh climatic conditions, such as high temperatures, cold, wind, and sand. In addition, the working range of an open-pit mine is large, the working surface often changes, the road conditions are relatively complex, the transportation distance is long, and the transportation task is heavy [1–6].

Open-pit mines are important sites for coal mining production and the main working sites for mining concrete mixing and transport tank trucks. However, the site usually contains

**Data Availability Statement:** All relevant data are within the manuscript and its Supporting Information files.

**Funding:** National Natural Science Foundation of China. Award Number: 52374123. Grant Recipient:

Guangwei LIU. The funder participated in the review and editing of this article.

many complex transportation routes and harsh and changeable weather conditions, and there are many slopes, so the safety issues during vehicle driving are very prominent [7]. The most prominent problem is the road condition. The road surface of an open-pit mine is usually uneven, including bumpy dirt roads, gravel roads, and ramps. These uneven road conditions will cause mining concrete mixing and transport tank trucks to produce severe bumps and vibrations during driving, thereby exacerbating the shaking of the concrete in the tank. Frequent vibrations and bumps will cause complex dynamic behaviors of the fluid inside the concrete, resulting in stratification, segregation, and even overflow of the concrete in the tank. For such problems, they can be solved by structural improvement or control method optimization. At present, there are adjustable advanced suspension systems [8, 9], intelligent transmission systems [10, 11], and remote diagnostic systems [12, 13] that can adapt to changing terrain conditions and have excellent performance in passing obstacles and slopes. Due to the limitations of dynamic response delay, insufficient adaptability to extreme terrain, large impact of load changes, and limited real-time intervention capabilities, the above solutions are still insufficient to solve the driving problems of mining concrete mixing tank trucks in complex environments such as uphill roads and sloped roads.

When mining concrete mixing tank trucks are driving on complex platforms, the sloshing of the fluid in the tank will cause non-axisymmetric problems, which greatly affects the dynamic characteristics of the vehicle body. When the vehicle is excited by complex road conditions, the fluid sloshing and the dynamic characteristics of the vehicle body will couple with each other, and in severe cases, it will cause accidents such as vehicle overturning. Therefore, it is crucial to analyze the stress conditions of mining concrete mixing tank trucks and optimize the control performance effects [14–16]. By deeply studying the stress conditions of vehicles under complex road conditions, the goal of this study is to solve the limitations of traditional solutions that are prone to overturning accidents when dealing with complex working conditions such as uneven roads and slopes by improving the adaptive hydraulic drive system. Fadhloun K et al. [17] developed a model to overcome this limitation by explicitly incorporating driver behavior into the mathematical expression of the dynamics-based acceleration model. The proposed model has a flexible shape, which enables it to incorporate driver changes. In addition, the model was shown to outperform similar models as it can predict more accurate acceleration levels in all areas. Xu Wenchao et al. [18] introduced the various components of a mining concrete mixer truck in detail through structural diagrams and cloud diagrams, providing a reference for a detailed study of mining concrete mixer trucks. To consider the trajectory tracking of UUSV and provide the best control strategy for ships with external disturbances, Karnani C et al. [19] proposed a controller based on model reference adaptive control with an integrator (MRACI) to ensure the stability of the closed-loop system. Due to external disturbances, the system response of the vehicle will change. The above scheme reduces the change to almost zero, thereby stabilizing the vehicle. Agbaje MB et al. [20] combined the improved firefly algorithm with the particle swarm optimization algorithm to solve the automatic data clustering problem. To investigate the performance of the proposed hybrid algorithm, its metaheuristic approach was compared using twelve standard datasets and two satellite datasets from the UCI Machine Learning Repository. Extensive computational experiments and result analysis show that the proposed algorithm not only achieves superior performance than the standard firefly and particle swarm optimization algorithms but also exhibits high stability and can be effectively used to solve other high-dimensional clustering problems. Khan A et al. [21] designed a novel and effective metaheuristic, population-based hybrid firefly particle swarm optimization (HFPSO) algorithm to solve different nonlinear and optimal power flow (OPF) problems. The results of the proposed algorithm were compared with the simulation results of the original particle swarm optimization (PSO) method and the most

advanced optimization techniques. The comparison of the optimal solutions shows that the proposed method can generate optimal, feasible, and global solutions with fast convergence speed, and can cope with the challenges and complexity of various OPF problems. Tian Meng-chu et al. [22] proposed a firefly-optimized particle filter algorithm based on the elastic mechanism to solve the particle depletion problem caused by the resampling of standard particle filters. The experimental results verified that the proposed algorithm can improve particle distribution, solve the particle depletion problem, and improve the comprehensive performance of particle filters. Sidorenko VS et al. [23] developed a general mathematical model of adaptive hydraulic drive with conveying control and obtained a new mathematical and computational model of the adaptive hydraulic drive using the speed control volume method of dual-mass dynamic system and hydraulic control circuit using the hydraulic mechanical multifunctional device. The process running in the hydraulic control circuit was identified, and the efficiency of the provided plate-oriented solution was proved. Wos P et al. [24] studied the structure, practical verification of identification, and control algorithm of an adaptive control electro-hydraulic servo system (EHSS) with external load disturbance, and calculated the computer program used to implement the algorithm and the numerical simulation and identification of the control physical model object. The research results show that the effectiveness of the adaptive control method in the electro-hydraulic servo system is verified from both theoretical and experimental aspects.

To improve the hydraulic drive control of the concrete mixer transport tank truck for mining, the firefly filter compensation optimization adaptive control algorithm [19, 25–27] uses the firefly algorithm to optimize the filter and control system parameters to achieve adaptive control. In this way, the system can maintain efficient and stable performance in complex environments. This paper adopts this method and introduces a method to design and implement flexibility and adaptability so that the system can respond and adjust quickly when dealing with changes and uncertainties to ensure continuous high performance and reliability of the elastic mechanism [28–30] to speed up the optimization process. The model of the optimized control system for the drive of the concrete mixer transport tank truck for mining was established, and the performance was compared with the traditional PID control method through comparative simulation analysis. Finally, the application effect of the optimized control in the open-pit mine transportation platform was evaluated.

## 2. Dynamic model of mining concrete mixing and transport tank truck

### 2.1 Mechanical model of the slope platform for mining concrete mixing and transport tank trucks

To study the motion state of a mining concrete mixing and transport tank truck when it is driving on a platform with a slope angle, the force analysis of the vehicle body is carried out, as shown in Fig 1, where $L_1$, $L_2$ the distances along the slope direction between the center of mass of the concrete tank truck and the contact points between the front and rear tires and the road slope, respectively, and $H$ is the average distance between the center of mass and the slope.

According to the static equilibrium equation, we can get:

$$F_{t1} + F_{t2} = mg \sin\theta \tag{1}$$

$$F_{n1} + F_{n2} = mg \cos\theta \tag{2}$$

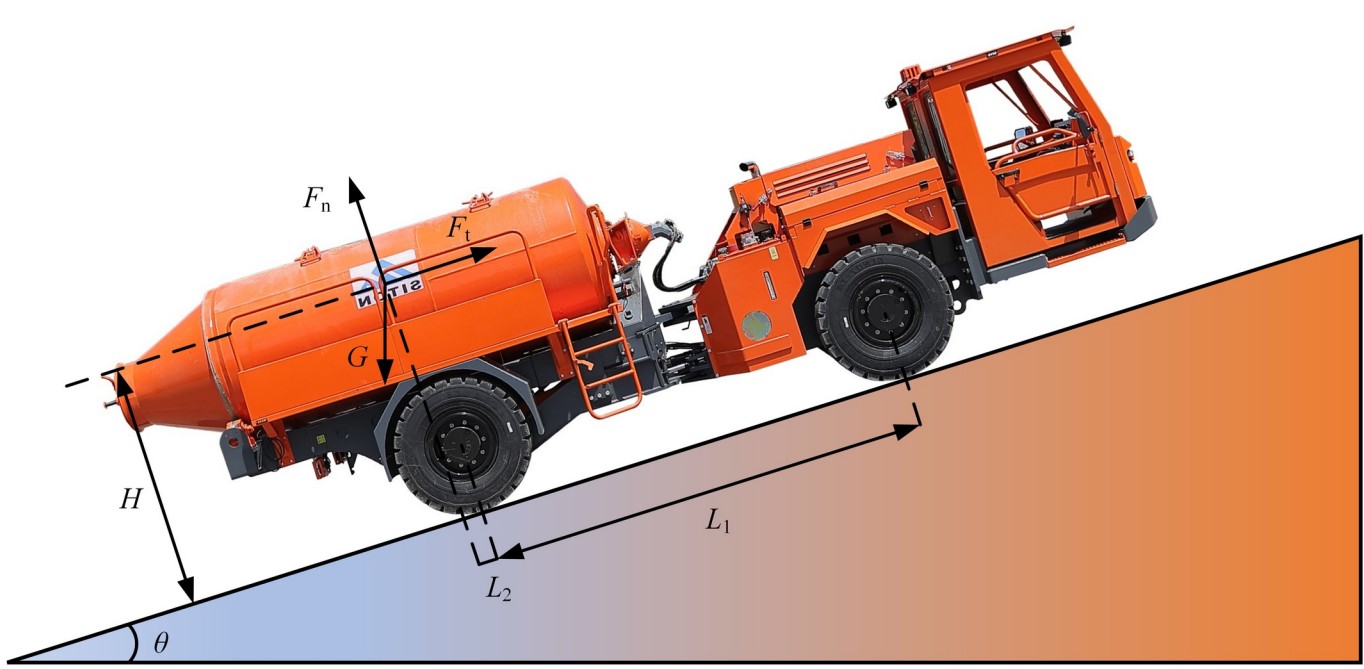

**Fig 1. Force diagram of KJCJ-4 mining concrete mixing tank truck driving on a sloping platform.**

Where: $F_{ti}$ is the vehicle traction/N; $F_{ni}$ is the vehicle positive pressure/N; $m$ is the mass of the transport tanker/kg; $g$ is the acceleration of gravity/(m/s$^2$); $\theta$ is the slope inclination.

The mining concrete mixing transport tanker is driving on the slope platform, and the interaction between the wheels and the slope is shown in Fig 2.

In the figure: $Z_i$ is the amount of tire axle sinking, $i$ = 1,2, corresponding to the front and rear wheels. $T$ is the torque generated when the wheel and axle are driving; $\omega$ is the angular velocity of the wheel; $v$ is the speed of the mining concrete mixing transport tank truck in the slope direction; $\alpha_1$ is the contact angle, $\alpha_2$ is the separation angle, $\alpha_3$ is the contact angle at the maximum load-bearing position, $\alpha_m = (c_1 + c_2 s)\alpha_1$. Among them: $\varphi_1$, $\varphi_2$ are the force

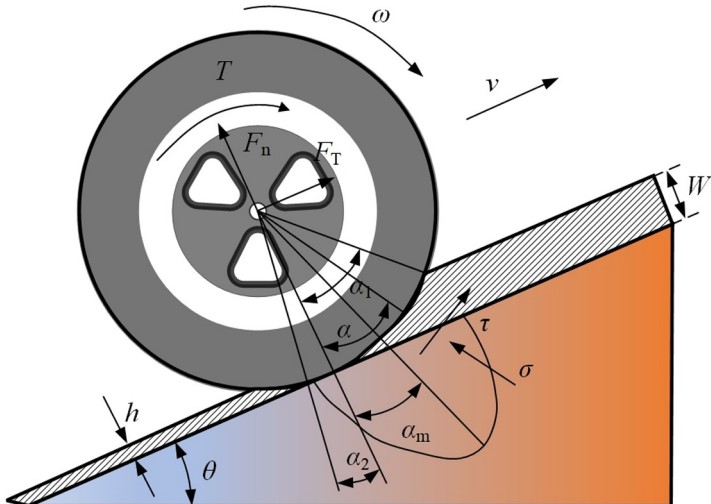

**Fig 2. Interaction model of tires of mining concrete mixing tank truck driving on a soft slope.**

coefficients of the sloping rock, and generally $c_1 = 0.4$, $0 \leq c_2 \leq 0.3$, $s$ are the wheel slip rate. $\tau$ represents the shear stress between the wheel and the slope, the normal stress is recorded as $\sigma$, and $h$ is the recovery amount after the elastic deformation of the slope platform.

The approach angle calculation formula is:

$$\alpha_1 = \arccos(1 - \frac{W_i}{r}) \tag{3}$$

Where $r$ is the tire radius/m.

The maximum stress point approach angle calculation formula is:

$$\alpha_{1\max} = (c_1 + c_2 s)\arccos(1 - \frac{W_i}{r}) \tag{4}$$

The normal stress calculation formula is:

$$\sigma = r^n(\frac{k_c}{w} + k_\varphi)(\cos\alpha_m - \cos\alpha_1)^n \tag{5}$$

Among them, $n$ is the deformation index, usually 1; $k_c$ is the slope rock change coefficient; $k_\varphi$ is the slope rock friction change coefficient; $w$ is the tire width/m.

According to the rock shear model [31, 32], the shear stress is:

$$\tau_m = (\varphi + \sigma\tan\theta) \times \left(1 - e^{-\frac{j(\theta)}{k}}\right) \tag{6}$$

$$j(\theta) = r[\alpha_1 - \alpha_m - (1 - s)(\sin\alpha_1 - \sin\alpha_m)] \tag{7}$$

According to the static equilibrium equation [33, 34], the normal pressure:

$$F_n = \frac{rw[\sigma(\alpha_1\cos\alpha_m - \alpha_m\cos\alpha_1 - \alpha_1 + \alpha_m) + \tau(\alpha_1\sin\alpha_m - \alpha_m\sin\alpha_1)]}{\alpha_m(\alpha_1 - \alpha_m)} \tag{8}$$

According to the static equilibrium equation, the traction force can be obtained:

$$F_t = \frac{rw[\tau(\alpha_1\cos\alpha_m - \alpha_m\cos\alpha_1 - \alpha_1 + \alpha_m) + \sigma(\alpha_1\sin\alpha_m - \alpha_m\sin\alpha_1)]}{\alpha_m(\alpha_1 - \alpha_m)} \tag{9}$$

Among them: $\varphi, k$ is the inherent shear characteristic parameter of rock, which represents cohesion/N and internal friction angle respectively; $\sigma$ is normal stress/N; $j(\theta)$ is shear displacement of soil/m; $\alpha_1$ is the entry angle of wheel thorn effect; $\alpha_m$ is wheel-ground action angle; $s$ is slip rate.

The wheel static subsidence model is used to calculate the contact pressure distribution between the wheel and the ground of the mining concrete mixing and transport tank, which helps to understand the force of the wheel under different loads and ground conditions. The model can evaluate the adaptability of the vehicle on different types of ground (such as uneven dirt roads, gravel roads ramps, etc.). By understanding the static subsidence characteristics of the wheel, you can choose a tire that is more suitable for specific road conditions to ensure the best driving performance of the vehicle under various road conditions.

Through the pressure model, the static sinking of the wheel can be calculated as:

$$Z_i = \left[ \frac{3Y}{bk\sqrt{2r}(3-n)} \right]^{\frac{2}{2n+1}}$$

(10)

$$k = k_c/b + k_\phi$$

(11)

Where: F is the wheel load; F is the coefficient; F represents a parameter related to the geometry, structure, or material properties of the tire; F represents a coefficient related to the stiffness, elasticity, or load-bearing capacity of the tire; F represents a variable related to the tire characteristics or usage conditions.

When a mining concrete mixer truck is driving on a slope, the rear tires often run in the track formed by the front tires. After the front tires have driven, the ground has been compacted. Therefore, when the rear tires are driving on the road surface where the front tires have driven, the deformation is greater. The subsidence coefficient [35] is a term used to describe the relative change in the deformation of the rear wheels when driving on the ground that has been compacted by the front wheels. To accurately represent the relationship between the front and rear wheel subsidence, the subsidence coefficient $\lambda$ is introduced,

$$\lambda = \frac{Z_1}{Z_2}$$

(12)

Where: $Z_1, Z_2$ represent the sinking amount of the front wheel, middle wheel, and rear wheel respectively.

The settlement ratio $\lambda$ is related to the properties of the slope platform rock mass and is obtained through a water pressure test. If there is more settlement, the slope is soft, the front wheel compaction effect is strong, the rear wheel settlement change is small, and the settlement is small. On the contrary, if there is less settlement, the slope platform rock mass is substantial, the front wheel compaction effect is weak, the rear wheel settlement change is also small, and the settlement ratio is small.

In summary, the mechanical model of a mining concrete mixing and transport tank truck driving on an open-pit mine slope can be expressed as:

$$\sum_{i=1}^{2} F_{mi} = \sum_{i=1}^{2} \frac{rw[\sigma_i(\alpha_{li}\cos\alpha_{mi} - \alpha_{mi}\cos\alpha_{li} - \alpha_{li} + \alpha_{mi}) + \tau_i(\alpha_{li}\sin\alpha_{mi} - \alpha_{mi}\sin\alpha_{li})]}{\alpha_{mi}(\alpha_{li} - \alpha_{mi})}$$
$$= mg\sin\theta$$

(13)

$$\sum_{i=1}^{2} F_{ii} = \sum_{i=1}^{2} \frac{rw[\tau_i(\alpha_{1i}\cos\alpha_{mi} - \alpha_{mi}\cos\alpha_{1i} - \alpha_{1i} + \alpha_{mi}) + \sigma_i(\alpha_{1i}\sin\alpha_{mi} - \alpha_{mi}\sin\alpha_{1i})]}{\alpha_{mi}(\alpha_{ii} - \alpha_{mi})}$$
$$= mg\cos\theta$$

(14)

## 2.2 Hydraulic drive system model

Most mining concrete mixing and transport tank trucks use a closed hydraulic drive system, and the structure is shown in Fig 3. Although this system has limitations such as system complexity, efficiency and energy loss, control accuracy, temperature influence, and environmental adaptability, it still has unique advantages in many applications, such as high power density

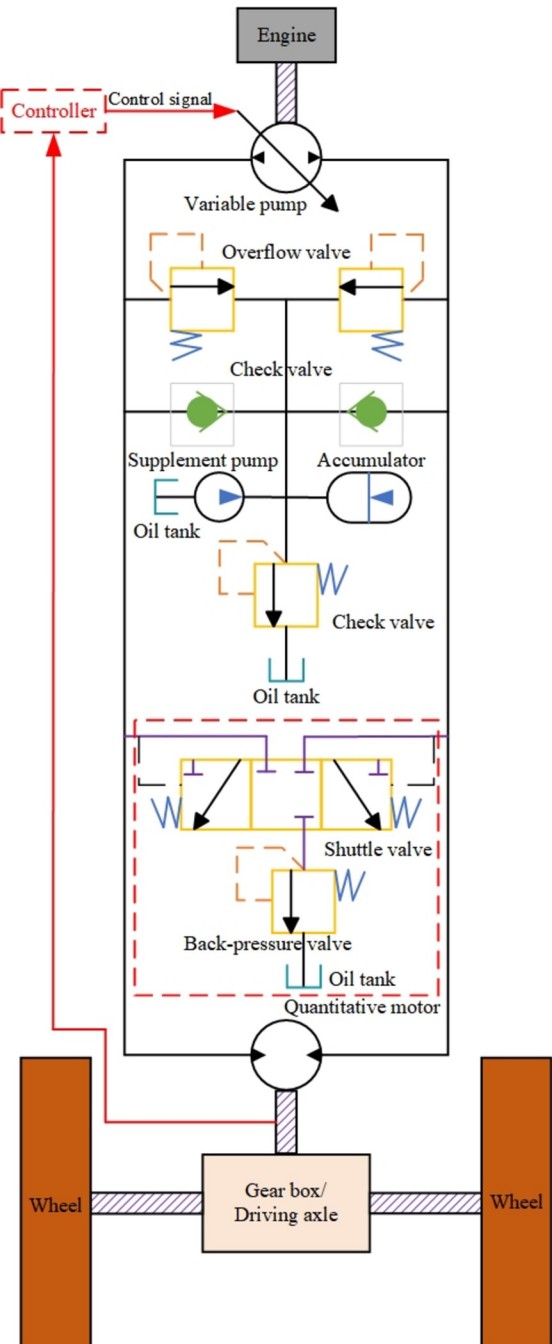

**Fig 3. Structure diagram of hydrostatic drive system.**

and flexible power transmission methods, so it is widely promoted and used in many industrial and engineering applications. The oil ports at both ends of the variable pump are connected end to end with the inlet and return ports of the quantitative motor to form a closed loop. The variable pump is driven by the engine to provide high-pressure oil to the system and can adjust the variable mechanism according to the control signal to achieve the speed and steering control of the quantitative motor. The system oil replenishment circuit consists of a one-way

valve, an accumulator, an oil replenishment pump, an oil tank, and a flow valve to replenish the leaked oil in time for the system to prevent cavitation. A pair of overflow valves provide overload protection for the system. In addition, the system flushing circuit consists of a shuttle valve, a back pressure valve, and an oil tank to exchange oil to prevent the system temperature from being too high.

As shown in Fig 3, assuming that the engine rotates at a constant speed, the variable pump output flow equation can be expressed as:

$$q_{\mathrm{P}} = k_{\mathrm{P}}\omega_{\mathrm{P}}u - C_{\mathrm{ip}}(p_1 - p_2) - C_{\mathrm{ep}}p_1 \tag{15}$$

Among them: $q_{\mathrm{P}}$ is the output flow of the variable pump/(m$^3$/s); $k_{\mathrm{P}}$, $\varphi_{\mathrm{P}}$, and $u$ are the displacement gain/(m$^3$/s), speed/(r/s), and control amount of the variable pump respectively; $C_{\mathrm{ip}}$ and $C_{\mathrm{ep}}$ are the internal and external leakage coefficients of the variable pump/(m$^3$/s/Pa); $p_1$ and $p_2$ are the inlet and outlet pressures of the quantitative motor respectively/Pa.

The effective elastic modulus of oil [36] is a key parameter that describes the volume change characteristics of oil in a hydraulic system when pressure changes. It reflects the combined effect of the compression of internal microbubbles and the compression of the oil itself when the oil is subjected to external pressure. It has an important impact on the dynamic response, control accuracy, and system stability of the hydraulic system. The effective elastic modulus of oil affects the compressibility of the oil, thereby affecting the solution of the flow continuity equation and the dynamic response of the system. The flow continuity equation of the quantitative motor can be derived as follows:

$$q_{\mathrm{M}} = C_{\mathrm{im}}(p_1 - p_2) + C_{\mathrm{em}}p_1 + D_{\mathrm{m}}\omega_{\mathrm{m}} + \frac{V_0}{\beta_{\mathrm{e}}}\dot{p}_1 \tag{16}$$

Among them: $q_{\mathrm{M}}$ is the motor input flow/(m$^3$/s), which is equal to the pump output flow; $C_{\mathrm{im}}$ and $C_{\mathrm{em}}$ are the motor internal and external leakage coefficients/(m$^3$/s/Pa); $D_{\mathrm{m}}$ is the motor displacement/m$^3$; $\omega_{\mathrm{m}}$ is the motor speed/(r/s); $V_0$ is the motor oil inlet volume/m$^3$; $\beta_{\mathrm{e}}$ is the effective elastic modulus of the oil/Pa.

Load torque [37] is an external torque acting on the rotating shaft or rotating element of a mechanical system, which directly affects the starting, stopping, stable operation, and energy consumption of the system. By accurately calculating and controlling the load torque, the operating efficiency and performance of the system can be improved, and the stability and reliability of the system under various working conditions can be ensured. The motor drives the gearbox through the output torque and then drives the tires through the drive axle. The traction and positive pressure of the concrete truck during driving are fed back to the motor through the tires as auxiliary torque. Therefore, the torque balance equation of the motor can be obtained as follows:

$$D_{\mathrm{m}}(p_1 - p_2) = J_t\dot{\omega}_{\mathrm{m}} + B_{\mathrm{m}}\dot{\omega}_{\mathrm{m}} + T_{\mathrm{L}} \tag{17}$$

Where: $J_t$ is the moment of inertia of the motor and drive axle/(kg·m$^2$); $B_{\mathrm{m}}$ is the rotation damping coefficient; $T_{\mathrm{L}}$ is the load torque/(N·m).

The motor drives the tire to rotate through the gearbox, so the speed of the mining concrete mixing tank truck is expressed as:

$$v = \omega_{\mathrm{m}}R_t i_{\mathrm{v}} \tag{18}$$

Where: $R_t$ is the tire radius/m, $i_{\mathrm{v}}$ is the speed ratio of the total gear.

Simultaneously, Eqs (15) and (18), with vehicle speed $v$ as the system output and variable pump control quantity $u$ as the system input, define the system state variable $x = [v, \dot{v}]$, and

the system state equation can be obtained as:

$$\begin{cases} \dot{x}_1 = x_2 \\ \dot{x}_2 = bu - a_1 x_1 - a_2 x_2 - d \\ y_1 = x_1 \end{cases} \quad (19)$$

Among them: $y_1$ is the system output; $b$ is the control gain; $a_1$ is the vehicle speed gain/(m/s); $a_2$ is the vehicle acceleration gain/(m/s$^2$); $d$ is the disturbance caused by the load during the wheel driving process/(N·m), which can be expressed as:

$$b = \frac{\beta_e D_m k_p R_t i_v}{V_0 J_t} \omega_p \quad (20)$$

$$a_1 = \frac{\beta_e C_t J_t + V_0 B_m}{V_0 J_t} R_t i_v \quad (21)$$

$$a_2 = \frac{\beta_c C_t B_m + \beta_c D_m^2}{V_0 J_t} R_t i_v \quad (22)$$

$$d = \frac{\beta_e C_t}{V_0 J_t} T_L + \frac{\beta_e D_m}{V_0 J_t} \dot{T}_L. \quad (23)$$

Among them, $C_t$ is the total leakage coefficient of the hydrostatic drive system, $C_t = C_{ip} + C_{ep} + C_{im} + C_{em}$.

## 3. Design of adaptive PID controller based on elastic FA algorithm optimization filter

The speed of the concrete mixing tank truck for mining is usually controlled by a PID control algorithm with a simple structure and high reliability. The PID control algorithm compares the actual vehicle speed with the reference vehicle speed to obtain the difference $e(t)$, and uses the PID coefficient $k_p, k_i, k_d$ to control the difference. Finally, a linear control law is output to control the concrete tank truck power system to achieve speed tracking control of the tank truck. The traditional PID control algorithm is shown in formula (24):

$$u(t) = k_p e(t) + k_i \int_0^t e(t) \mathrm{d}t + k_d \frac{\mathrm{d}e(t)}{\mathrm{d}t} \quad (24)$$

However, when driving on a complex slope platform for a long time, the mining concrete mixing tank truck needs to withstand the influence of the rotating torque of the mixing drum and the change of the center of mass during transportation. The traditional PID control algorithm makes it difficult to meet the requirements of driving stability, high-speed accuracy, and strong robustness of mining concrete mixing tank trucks under complex working conditions, especially when facing the significant impact of fluid sloshing on vehicle dynamics. Fluid sloshing can cause dynamic changes in vehicle load, affecting the stability and control accuracy of the vehicle. The traditional PID control algorithm shows obvious limitations in dealing with such complex dynamic load changes and cannot provide sufficient response speed and adjustment accuracy. To solve the above problems, this paper proposes an FA-optimized particle filter adaptive PID controller based on an elastic mechanism, that is, an exponentially convergent disturbance observer is used to estimate the system disturbance and unmodeled

error, and the system dynamics are compensated in combination with the nominal model to achieve disturbance suppression and improve system robustness.

## 3.1 FA optimization principle of elastic mechanism

This method uses the attraction and movement mechanism of FA to optimize particle distribution, further designs the particle interaction mode establishes the elastic optimization strength criterion, prevents unreasonable particle aggregation and over-optimization from causing computational burden and low-quality convergence, accelerates the adjustment of PID parameter optimization speed, and is more suitable for parameter optimization of adaptive control.

### 3.1.1 Gravity and movement mechanisms.

*(1) Particle fluorescence brightness.* The latest measurement value is introduced to measure the position of the particle. The particle fluorescence brightness formula is defined as

$$I_n^i(t) = I_0 e^{\left[-\gamma z(t) - z_n^i(t)\right]} \tag{25}$$

$\gamma_n^i(t)$ is the fluorescence brightness of particle $\gamma$ in the Nth iteration at time $t$, $\gamma_0$ is the initial value of the fluorescence brightness of particle $\gamma$, $z(t)$ is the latest measured value of the filter in the iteration at time $t$, and $\gamma_n^i(t)$ is the predicted value of particle $\gamma$ in the $n$th iteration at time $t$.

The meaning of the fluorescence value is to judge the position of a particle. The higher the fluorescence value of a particle, the better the position of the particle. In each iteration, the particle with the highest fluorescence value is defined as the optimal particle $\gamma_{max}$ in this iteration.

$$\gamma_{max}(t) \in \left\{x_n^i(t), i = 1, 2, ..., N | \gamma(x)\right\} = \max\{\gamma[x_n^i(t)], i = 1, 2, \ldots, N\} \tag{26}$$

Where: $\gamma_{max\ n}(t)$ is the optimal particle state value in the $n$th iteration at the $t$th time; $\gamma_n^i(t)$ represents the state value of particle $\gamma$ in the $n$th iteration at the $t$th time; $\gamma(\cdot)$ represents the calculation of fluorescence brightness.

*(2) Particle attraction.* Particles need to use the best particle to guide the movement of other particles. The gravitational force between the best particle $\gamma_{max}$ and particle $\gamma$ directly affects the distance that particle $\gamma$ moves to the best particle $\gamma_{max}$. The formula for the attraction of the best particle $\gamma_{max}$ to particle $\gamma$ is:

$$\beta_n^i(t) = 1 - \beta_0 e^{-\gamma r_{i\gamma_{max}n(t)}^2} \tag{27}$$

Where: $\beta_n^i(t)$ represents the attraction of the optimal particle $\gamma_{max}$ to particle $\gamma$ in the $n$th iteration at the $t$th time, $\beta_0$ represents the initial value of the attraction of the optimal particle $\gamma_{max}$ to particle $\gamma$, and $r_{i\gamma_{max}n(t)}$ is the spatial distance between particle $\gamma$ and the optimal particle $\gamma_{max}$.

*(3) Particle position update.* All particles' initial positions are randomly distributed in the search space. The distribution is updated by guiding other particles to move with the best particle. The particle position update is shown in formula (26):

$$x_m^i(t) = x_{n-1}^i(t) + \omega_2[\text{gbest}_n(t)] + \omega_1(\text{rand} - \frac{1}{2}) \tag{28}$$

Where: $x_n^i(t)$ is the state value of particle $\gamma$ after the $n$th iteration at time $t$, $x_{n-1}^i(t)$ is the state value of particle $\gamma$ after the $n$-1th iteration at time $t$, and $\gamma_{max n}(t)$ is the optimal point of the $n$th iteration at time $t$; $\omega_1$ and $\omega_2$ represent the gain coefficients of the optimal particle and random

function respectively, and rand represents a random number between [0,1) that satisfies a uniform distribution.

**3.1.2 Elasticity optimization strength control.**   The proportion of high-likelihood particles describes the particle distribution, the optimization intensity is controlled by the maximum number of iterations, and the particle density is set. The detection mechanism controls the particle density around the optimal particle to improve the particle distribution so that most particles are concentrated in the high-likelihood area, and particles are distributed in other areas.

*(1) Particle density*. First, the particle density is defined, and a quantitative indicator is set to determine whether the particles are too dense. After the position update is completed in the $n$th iteration, most particles move to the optimal particle. With the optimal particle in the $n$th iteration as the center, the judgment radius is $r_{judg}$, and the number of particles in the judgment area is defined as the particle density $cr_m$.

$$cr_m = N\left\{x_n^i(t) - \gamma_{max\ n}(t) \leqslant r_{judg}\right\} \tag{29}$$

Where: $x_n^i(t)$ is the position of the particle $i$ after movement in the $n$th iteration at time $t$, $\gamma_{max\ n}(t)$ is the optimal particle position in the $n$th iteration at time $t$, and $N$ represents the calculation of the number of particles in the judgment area.

*(2) Ratio of high probability particles [38]*. The ratio of high-likelihood particles is defined. The optimization strength of particles is controlled by combining the ratio of high-likelihood particles and the maximum number of iterations, which is defined as

$$pr_m = \frac{cr_m}{N} \tag{30}$$

When the ratio of high-likelihood particles in the $n$th iteration is greater than the set maximum ratio $pr_{max}$ of high-likelihood particles or the number of iterations is greater than the maximum number of iterations, the optimization is stopped.

**3.1.3 Particle density detection.**   To measure the particle distribution after iteration, if the high likelihood particle ratio calculated in real-time exceeds the set threshold $pr_{max}$ (should be slightly higher than the maximum high likelihood particle ratio), it means that the particles around the optimal particle are too dense and the particle distribution around the optimal particle needs to be adjusted. A spring system is constructed between the particles in the judgment area and the optimal particle. The compression of the spring is equal to the distance the particle moves to the optimal particle. When the particles are too dense, the spring will cause the particles to bounce in the opposite direction of the compression, increasing the compression.

*(1) Rebound coefficient*. In the $n$th iteration, the particle moves to the optimal particle, and the moving distance is the compression of the spring, which is defined as

$$\Delta x_n^i = x_n^i(t) - x_{n-1}^i(t) \tag{31}$$

According to Hooke's law [39], the corresponding elastic force is:

$$F_n^i = k_s \Delta x_n^i \tag{32}$$

Where $k_s$ is the elastic coefficient of the interparticle spring and the rebound coefficient of the rebounding particle is defined as:

$$\zeta = e^{\left(-\frac{c}{F_n^i}\right)} \tag{33}$$

Among them, $c$ is a constant, and the rebound distance should equal the distance the particle moves. Therefore, the constant c is set to adjust the rebound coefficient, and the rebound coefficient can be obtained by substituting it into formula (30) in formula (31).

$$\zeta = e^{[-\frac{c}{k_s}/\Delta x_n^i]} \tag{34}$$

It can be seen that $\frac{c}{k_s}$ is a constant, the value of $k_s$ is related to the coefficient of rebound, and the influence of the value of $k_s$ can be eliminated by $c$, so for the convenience of calculation, the coefficient of rebound can be directly taken as 1.

*(2) Update of particle position under the action of an elastic mechanism.* It is determined that the particles in this area rebound in the opposite direction of compression. The rebound distance is affected by the rebound coefficient. The position of the rebounded particle is updated as follows:

$$\hat{x}_n^i(t) = x_n^i(t) + \zeta[x_{n-1}^i(t) - x_n^i(t)] \tag{35}$$

## 3.2 Particle weight filter compensation control design

After the FA algorithm is optimized, the particle state changes, and the distribution density function represented by the particle subset is no longer $p[x(t)|Z(t-1)]$. Therefore, compensation measures must be taken to make the particle set with weights before and after optimization theoretically obey the same distribution $p[x(t)|Z(t-1)]$. This paper adopts the idea of necessary sampling and the definition formula of weights to filter and compensate for the optimized particle weights. The weight compensation process here differs from the particle weight solution process in formula (2), but the uniform distribution process of weighted particles before and after optimization.

Assume that the posterior probability density function $p[x(t)|Z(t-1)]$ is represented by the particle set $\{x^i(t-1), \omega^i(t-1)\}_{i=1}^N$ at time $t$-1, and the one-step prediction density function $p[x(t)|Z(t-1)]$ is represented by the new prediction particle set $\{\tilde{x}^i(t)\}_{i=1}^N$ at time $t$, represented by the improved optimized particle set. In other words, the optimized sample set $\{x^i(t)\}_{i=1}^N$ can be regarded as a sample set from the probability density function $g[x(t)]$, rather than a sample set from the probability density function $p[x(t)|Z(t-1)]$. Therefore, when the particle position changes, the weight should be filtered and compensated to ensure the uniformity of the weighted particle distribution before and after optimization. Take $g[x(t)]$ as the importance density function of $p[x(t)|Z(t-1)]$. According to the definition of weights:

$$R^i(t) = \frac{p[x^i(t)|Z(t-1)]}{g[x^i(t)]} \tag{36}$$

Combined with formula (36), the weight compensation function of the optimized particle is defined as:

$$\omega^i(t) \propto \omega^i(t-1)R^i(t)p[z(t)|x^i(t)] \tag{37}$$

The specific design method is as follows:

Combined with formula (19), The perturbation observation dynamics [40] is defined as:

$$\dot{\hat{d}} = K(d - \hat{d}) = K(bu - a_1 x_1 - a_2 x_2 - \dot{x}_2) - K\hat{d} \tag{38}$$

Where $\hat{d}$ is the estimated value of the system disturbance; $K$ is the disturbance convergence coefficient, $K > 0$.

Define the auxiliary function:

$$z = \hat{d} + Kx_2 \tag{39}$$

Then, the dynamic equation of Eq (39) obtained by combining Eq (38) is:

$$\dot{z} = \dot{\hat{d}} + K\dot{x}_2 = K(\mathrm{b}u - a_1 x_1 - a_2 x_2) - K\hat{d} \tag{40}$$

Therefore, combining Eqs (39) and (40), the exponentially convergent disturbance observer is designed as:

$$\begin{cases} \dot{z} = K(bu - a_1 x_1 - a_2 x_2) - K\hat{d} \\ \hat{d} = z - Kx_2 \end{cases} \tag{41}$$

The mining concrete mixing tank truck is mainly subject to slow interference during driving, so it can be assumed that the dynamic $\dot{d} = 0$ of the actual interference is:

$$\dot{\tilde{d}} = \dot{d} - \dot{\hat{d}} = -\dot{\hat{d}} = -\dot{z} + K\dot{x}_2 \tag{42}$$

Where $\tilde{d}$ is the difference between the actual disturbance and the observed disturbance. Combining Eqs (39) and (40), we get

$$\dot{\tilde{d}} = -K(bu - a_1 x_1 - a_2 x_2) + K\hat{d} + K\dot{x}_2 = (d - \hat{d}) = -K\dot{\tilde{d}} \tag{43}$$

Therefore, the system disturbance observation error equation can be obtained as follows:

$$\dot{\tilde{d}} + K\dot{\tilde{d}} = 0 \tag{44}$$

Then the solution of Eq (41) is:

$$\tilde{d}(t) = \tilde{d}(t_0)e^{-Kt} \tag{45}$$

That is, at time $t \to \infty$, the disturbance observation error tends to 0, and its convergence speed is positively correlated with the gain $K$. The larger the $K$ value, the faster the convergence speed; but too large a $K$ value will introduce too much noise, so it is necessary to compromise the $K$ value.

Formula (41) shows that the observer can effectively observe the system disturbance and compensate for the disturbance in the PID controller. At the same time, combined with the model (19) to compensate for the system dynamics, the PID controller based on the

disturbance observer can be designed as

$$u = \frac{k_p e + k_I \int_{t_0}^{t} e \mathrm{d}t + k_D \frac{\mathrm{d}e}{\mathrm{d}t} + \ddot{y}_d + a_1 x_1 + a_2 x_2 + \hat{d}}{b} \qquad (46)$$

Where: $e$ is the vehicle speed tracking, defined as $e = y_d - y$, $y_d$ is the target tracking curve; $k_p$, $k_1$ and $k_D$ are the proportional, integral, and differential gains of the elastic firefly optimization filter adaptive PID controller; $b$ is the control gain.

To prove the convergence of the proposed controller, after optimization compensation, Eq (42) is substituted into Eq (19) to obtain

$$k_p e + k_1 \int_{t_0}^{t} e \mathrm{d}t + k_D \dot{e} + \ddot{e} - \tilde{d} = 0 \qquad (47)$$

From formula (41), it can be seen that $\tilde{d}$ converges exponentially. Therefore, by adjusting the control gains $k_p$, $k_1$, and $k_D$, the error term $e$, the error integral term, and the error dynamic term can converge exponentially, and the convergence accuracy is jointly determined by the four gains $k$, $k_p$, $k_1$, and $k_D$ k. A linear differentiator obtains the disturbance of the desired signal and the feedback signal. The differentiator is designed as follows:

$$\dot{\hat{v}}_1 = \hat{v}_2$$
$$\dot{\hat{v}}_2 = \hat{v}_3$$
$$\dot{\hat{v}}_3 = R_2^3 [-c_1(\hat{v}_1 - v_1) - c_2 \frac{\hat{v}_2}{R_2} - c_3 \frac{\hat{v}_3}{R_2^2}] \qquad (48)$$

Where: $v_1$ is the original signal; $\hat{v}_1$ is the estimated value of the original signal $v_1$; $\hat{v}_2$ is the estimated value of the first-order differential of $v_1$; $\hat{v}_3$ is the estimated value of the second-order differential of $v_1$, $c_i$ is an array that satisfies Hurwitz, $c_1 = 1$, $c_2 = 3$, $c_3 = 3$; $R$ is the adjustment coefficient, $R = 50$ in this article.

In summary, the elastic FA filter optimization adaptive PID control structure diagram of the static hydraulic drive system of the concrete tank truck is shown in Fig 4.

## 3.3 Optimizing PID parameters using the elastic firefly algorithm

Many algorithms can optimize PID at present. Although they have the characteristics of solving problems, their limitations are also particularly obvious. Genetic algorithms can effectively explore the parameter space and have a high probability of finding the global optimal solution. They are suitable for complex and multi-peak optimization problems and do not require the gradient information of the objective function. However, the computational complexity is high, especially when the parameter space is large. The convergence process may be slow and require multiple iterations. Algorithm parameters such as population size, crossover rate, and mutation rate need to be set. Particle swarm optimization algorithm has the advantages of fast convergence speed, simple algorithm, easy implementation, low requirements in the form of objective function, and applicability to a variety of problems. However, in a complex search space, it may fall into a local optimal solution, and parameters such as the number of particles and speed limit need to be set.

After establishing the elastic FA algorithm optimized filter adaptive PID control structure of the hydrostatic drive system of a mining concrete mixer transport tank truck, the elastic FA algorithm can globally optimize the PID control parameters to ensure that the system can maintain optimal performance under various working conditions. By adjusting the control parameters in real-time, the system can achieve more precise control and reduce overshoot

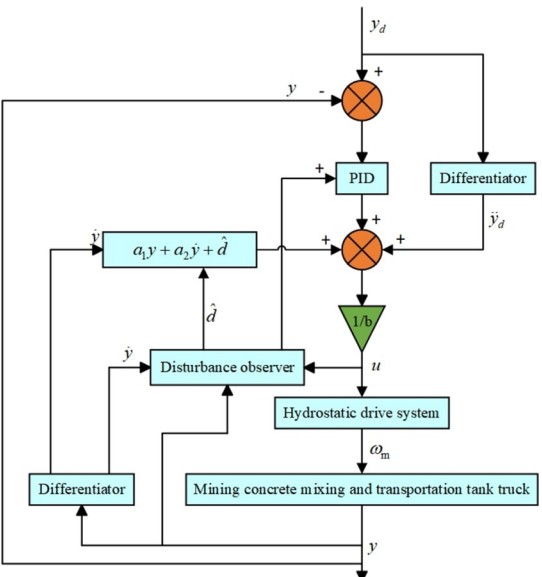

**Fig 4. Structure diagram of elastic FA algorithm optimized filter adaptive PID control.**

and oscillation. The elastic FA algorithm has good global search ability and adaptability and can find the optimal control parameters under different working conditions. Whether under high load or low load conditions, the system can maintain efficient operation through adaptive adjustment. At the same time, its strong adaptability, modular design, real-time, and integration enable it to maintain efficient and stable operation under various complex working conditions, especially in real-time control applications, which can significantly improve the system's response speed, control accuracy, and stability. Next, the elastic firefly algorithm is used to optimize the PID parameters. The specific optimization steps are as follows:

1. Determine the values of the parameters in the PID controller according to the engineering method [41]:

range $k_p \in [0, 100], k_i \in [0, 1], k_d \in [0, 1]$, select the initial population size $N = 50$, the initial fluorescence brightness $I_0 = 5$, the initial value $\beta = 0.4$ of the optimal particle attraction, the gain coefficients $\omega_1$ and $\omega_2$ of the optimal particle and random function are 0.09 and 0.05 respectively, the judgment radius $r_{judg} = 0.04$, the maximum ratio of high-likelihood particles $pr_{max} = 0.95$, and the maximum number of iterations is 100.

2. Iterate according to the particle attraction and particle density calculation in 3.2.

3. Calculate whether the cutoff condition is met. If the cutoff condition is met (the maximum number of iterations or the ratio of high-likelihood particles is greater than the maximum ratio $Pr_{max}$), the optimal solution is output.

Through the above steps, the initialization result optimization process first reached the cutoff condition of the maximum ratio $Pr_{max}$ of high-likelihood particles exceeding 0.95 in step 67. The PID parameter tuning result after optimization by the elastic FA algorithm is: $k_p = 47.832, k_i = 0.718, k_d = 0.697$.

## 4. Establishment and verification of the simulation model

### 4.1 Simulation model establishment

According to the physical characteristics of the mining concrete mixing tank truck, the system state variable $x = [\omega_1, \beta_1, \frac{d\varphi_1}{dt}, \frac{d\varphi_2}{dt}, \frac{d\theta}{dt}, \theta, \varphi_1, \varphi_2]^T$ is selected, which can be written in the form of a matrix differential equation [18, 42]:

$$M\frac{dx}{dt} = Cx + N\delta + w \tag{49}$$

In the formula, $M$, $C$, and $N$ are coefficient matrices; $w$ is the additional term generated by the relative motion of each order of mass in the equivalent system relative to the tank.

Then, the state space equation of the system can be described as:

$$\frac{dx}{dt} = Ax + Bu + Ew \tag{50}$$

In the formula, $A \in R^{8\times8}$ is the system matrix, and $A = M^{-1}C$; $B \in R^{8\times1}$ is the input matrix, and $B = M^{-1}N$; $u = \delta$ is the system input; $E = M^{-1}$.

The dynamic simulation model of the mining concrete mixing and transport tank truck is established, as shown in Fig 5.

### 4.2 Control simulation system experiment

This study uses the R2023a Win64 version of MATLAB software, running on Windows 11 system, and the processor is Intel(R) Core(TM) i7-14700HX 2.10 GHz.

MATLAB has powerful numerical simulation capabilities and has been widely used in research fields such as vehicle dynamics, mechanical kinematics, and control. Simulink is a graphical programming environment provided by MATLAB for modeling, simulating, and analyzing dynamic systems. It uses graphical block diagrams to build and connect system models, each block representing a function or operation. The entire model architecture is divided into dynamic subsystems, liquid sloshing subsystems, controller subsystems, environmental subsystems, data acquisition and recording subsystems, etc. This paper uses the

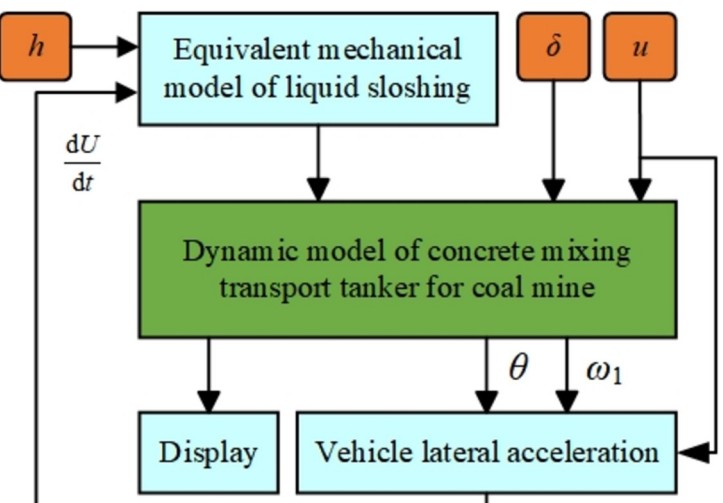

**Fig 5. Dynamic simulation model of mining concrete mixing and transport tank truck.**

MATLAB/Simulink simulation tool library [43, 44] to establish a multi-factor simulation model that integrates the tank truck dynamics model, hydrostatic drive system model, rock slope model, and elastic FA algorithm optimization filter adaptive PID controller model, and uses Simscape and Vehicle Dynamics Toolbox for multi-physics field modeling and vehicle dynamics analysis. Simscape is used for fluid dynamics simulation, and Vehicle Dynamics Toolbox is used for vehicle model establishment and simulation. As shown in Fig 6.

In the controller module of the simulation model, this paper integrates two control algorithms: traditional PID and elastic mechanism FA algorithm optimized particle filter PID control and uses these two algorithms to perform stable control on the concrete tank body. Among them, the elastic FA algorithm optimized filter adaptive PID control mechanism block diagram is shown in Fig 5. To verify the effect of the proposed elastic FA optimized filter adaptive PID control algorithm, simulation experiments of the two algorithms were carried out.

To verify that the proposed elastic FA algorithm optimized filter adaptive PID control has the performance of limiting control error and reducing control overshoot, a step speed signal with an amplitude of 5km/h is first given to the controller at time $t = 1$s, and hydraulic drive control is performed through traditional PID control and elastic FA algorithm optimized filter adaptive PID control. The experimental results are shown in Fig 7.

As shown in Fig 7, under the step signal, the elastic FA optimized filter adaptive PID control algorithm can achieve a stable motion state at time $t = 5.4$s, the steady-state error is controlled within 0.01km/h, and the tanker speed has no overshoot. Compared with the traditional PID control, the stabilization time is advanced by about 1.6s, and the speed stability control accuracy is improved by about 0.03km/h. The simulation proves that the indicators of the proposed control algorithm are better than those of the traditional hydraulic drive control algorithm.

At the same time, to prove that the proposed elastic FA algorithm optimized filter adaptive hydraulic drive control strategy has strong robustness when driving on soft mud slopes after the speed reaches stability, the tanker is allowed to drive onto a platform road with an angle of 20˚ to test the control effect of the soft slope road. The simulation data is shown in Fig 8.

When time $t = 10$s, the transport tanker drives onto a platform road with a slope of 20˚; when $t = 16$s, the transport tanker drives off the slope platform. According to the analysis of Fig 7, the traditional PID control strategy is used when the tank truck drives onto the 20˚ platform road. The speed of the tank truck is affected by the partial gravity along the slope, and a

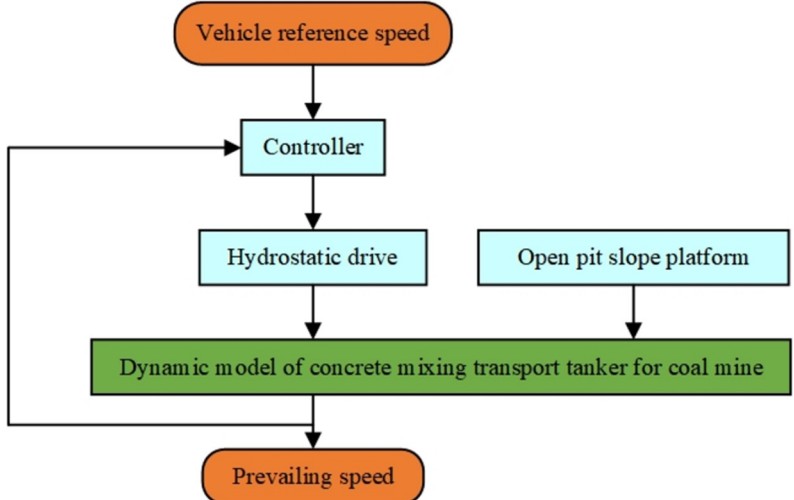

**Fig 6. Simulation model of mining concrete mixing tank truck driving on rock slope.**

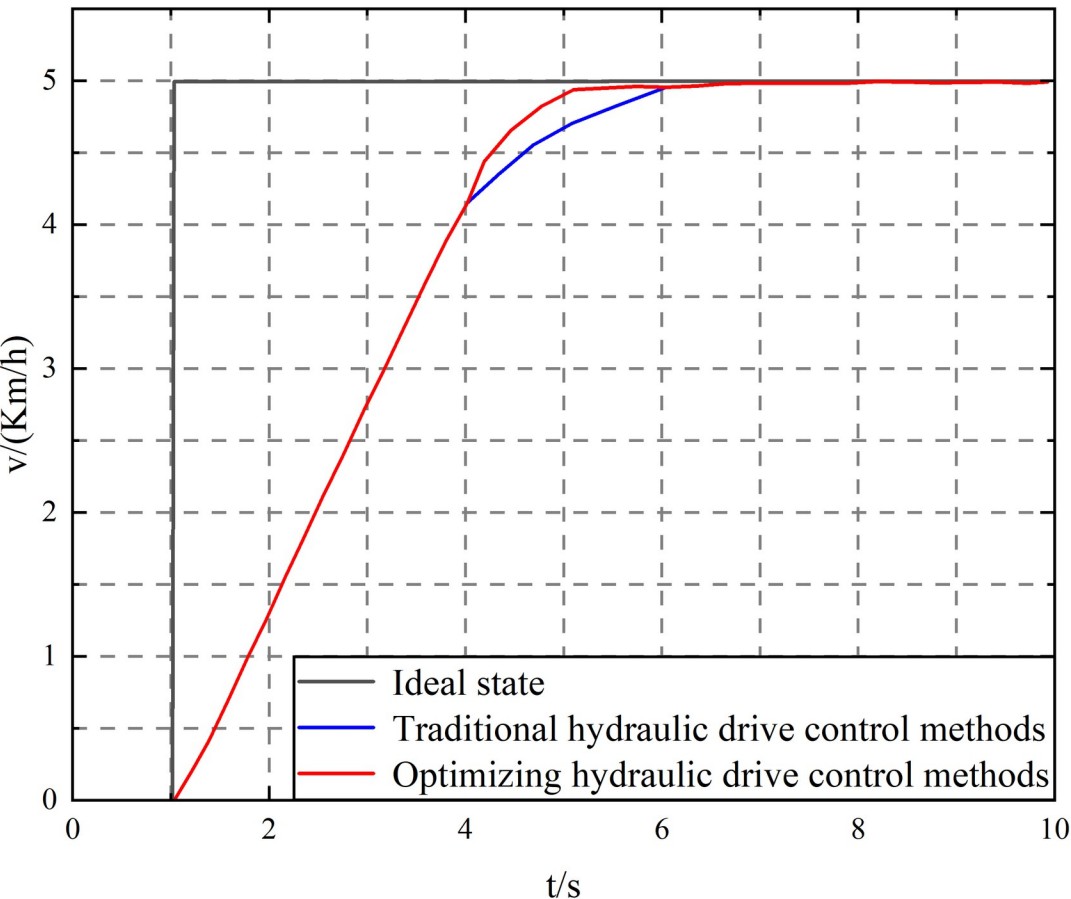

**Fig 7. Speed control of mining concrete mixing tank truck under step signal.**

certain degree of speed drop and a steady-state error of about 0.22 km/h appears. The steady-state error gradually decreased, and the flat ground driving state was restored when the tank truck left the ramp. When the elastic FA optimized filter adaptive PID control algorithm with stronger robustness is used, the tank truck only generates a transient error of about 0.04 km/h when driving on and off the slope. The speed error can converge quickly under the control of the nonlinear control law and return to a stable driving state. It can be seen that even on the soft mud slope, the proposed algorithm can still control the tank truck's speed in a stable driving state and effectively ensure the speed tracking control accuracy of the tank truck. Compared with the traditional PID control, the elastic FA-optimized filter adaptive PID control algorithm has stronger robustness.

The speed-tracking performance of concrete mixer trucks for mining [45] has an important impact on the safety and production efficiency of open-pit mine production operations. Accurate speed tracking performance ensures that the vehicle can travel at the predetermined speed and avoids sudden acceleration or deceleration, thereby improving the stability of operation. In the complex and dangerous environment of the mining area, accurate control of vehicle speed can reduce the risk of collision and rollover. By optimizing the control algorithm, regular maintenance, and operator training, the speed tracking performance can be effectively improved, thereby improving the safety and production efficiency of operations. In actual operation, stable speed control can not only reduce the risk of accidents but also improve

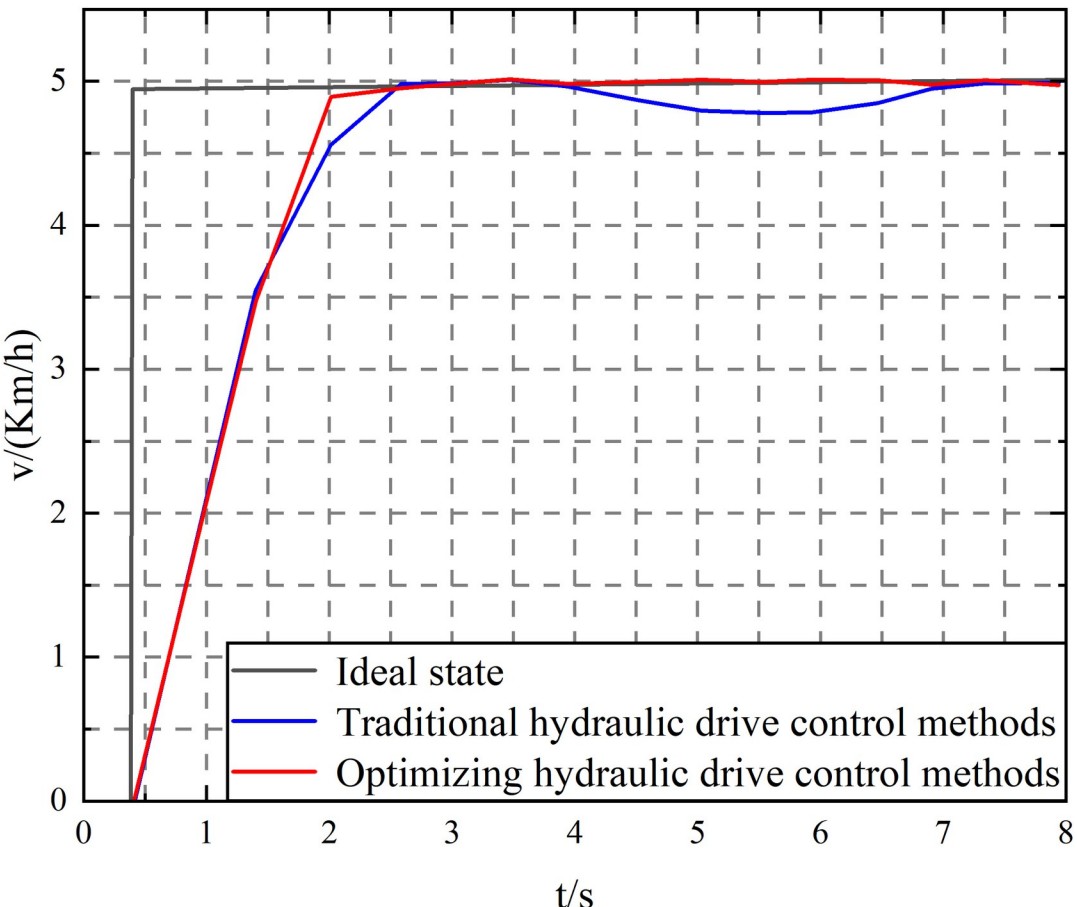

**Fig 8. Comparison of ramp driving performance of mining concrete mixing tank trucks.**

transportation efficiency and concrete quality, ensuring the smooth progress of open-pit mine production operations.

White noise [46] refers to noise whose power spectrum density is constant in the entire frequency domain. Random noise with the same energy density at all frequencies is called white noise. White noise has the characteristics of uniform spectral distribution and randomness and has a wide range of effects on the control system, vibration and noise, concrete mixing quality, equipment failure and maintenance, and driving of concrete mixer trucks for mining. By enhancing the robustness of the control system, improving the vibration and sound insulation of the vehicle, regular maintenance and inspection, and training operators, white noise interference can be effectively dealt with and the performance and stability of the vehicle can be improved.

To comprehensively compare the control performance of the tank truck hydraulic drive by the traditional PID and the elastic FA algorithm optimized filter adaptive PID control, this paper simulates the uniform speed and acceleration and deceleration driving control on a flat road and a 20° platform road and adds white noise interference to the feedback signal to verify the robustness of the proposed method under noise interference. Among them, when time = 75s, the tank truck drives onto the platform road with a slope of 20°; when = 275s, the tank truck drives off the slope platform. The total simulation time is 350s, the speed error is,

and the tank truck driving speed tracking and speed tracking control error result data are shown in Fig 10 respectively.

The test results are analyzed, as shown in Fig 9A and 9B. The tank truck speed control performance using the elastic FA optimized filter adaptive PID control algorithm is better, exceeding the traditional PID control algorithm. Throughout the experiment, the average speed control error of the traditional PID control algorithm was 0.13km/h, the maximum speed error was 0.37km/h, and the root mean square error was 0.12; while the elastic FA-optimized filter adaptive PID control had an average speed control error of 0.03km/h, a maximum speed error of 0.17km/h, and a speed error root mean square of 0.07. Moreover, as shown in Fig 8C, the disturbance observer optimized by the FA algorithm can accurately observe the system disturbance and realize accurate compensation for the system disturbance, thereby improving the system's anti-interference ability. The elastic FA-optimized filter adaptive PID control can significantly improve the speed control accuracy and robustness of the mining concrete mixer truck. This control strategy not only improves the control accuracy of the concrete mixer truck under various operating conditions but also enhances the robustness of the system, enabling it to operate stably in complex environments. Through these improvements, the overall efficiency and reliability of the concrete mixer truck have been significantly improved, which is manifested in improved production efficiency, reduced failure rate, reduced maintenance costs, and extended equipment service life. These benefits are of great

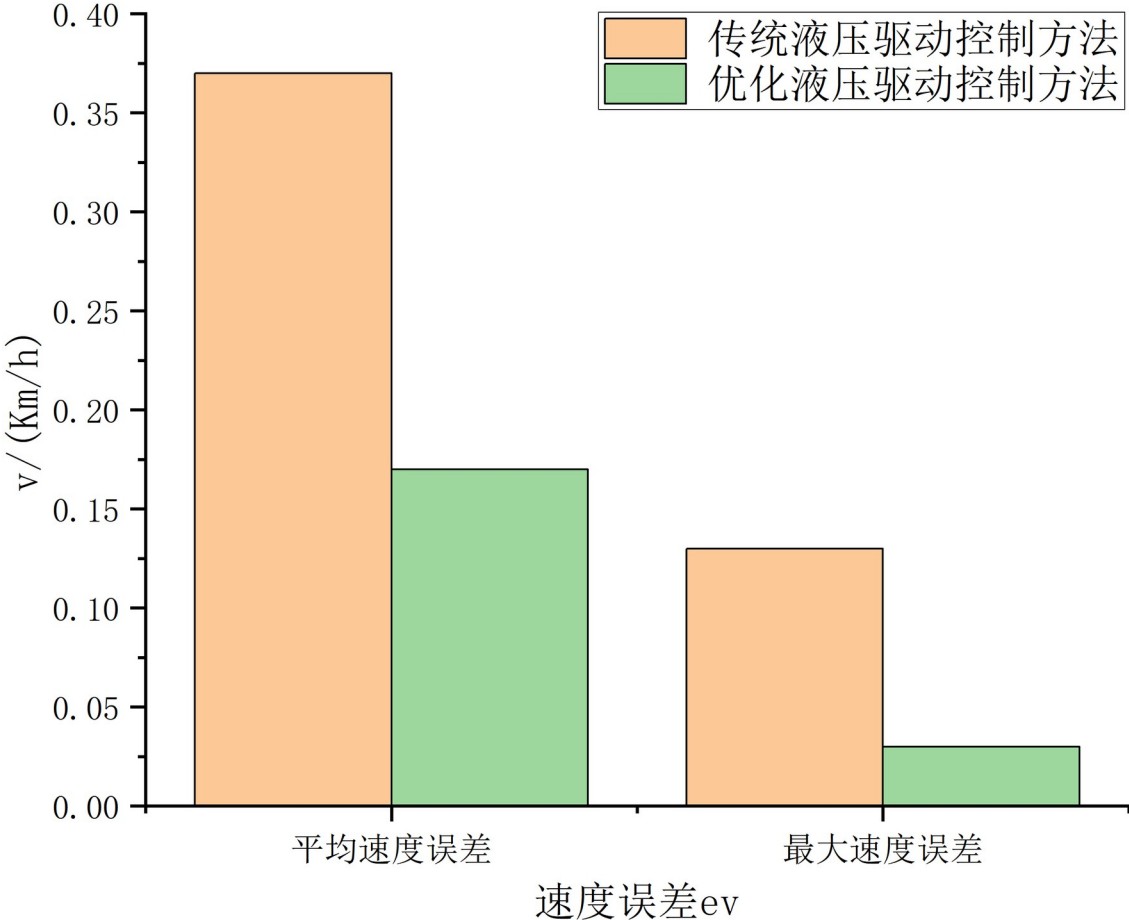

**Fig 9. Comparison of speed errors between traditional and improved hydraulic drive control methods.**

significance to improving the overall operational efficiency of mining concrete mixer trucks and reducing operating costs. Therefore, the optimized control model is better than the traditional PID control model.

The fluid sloshing disturbance characteristics and modeling of mining concrete mixing tank trucks involve aspects such as fluid dynamics, vehicle dynamics, and control system modeling. The model can accurately describe the behavior of fluid sloshing, which has an important impact on the stability of the tank truck, the quality of concrete, and the design and performance of the control system. By using physical modeling, numerical simulation, and control system modeling, the challenges brought by fluid sloshing can be better addressed, and the overall efficiency and reliability of mining concrete mixing tank trucks can be improved. Dynamic disturbances simulating fluid sloshing in the tank are added to the simulation, and constant speed acceleration and deceleration driving control simulation tests are carried out on flat roads and continuous roads with a soft slope of 20˚ to verify the robustness of the elastic FA algorithm to optimize the filter adaptive PID control. The tank truck driving tracking speed and speed tracking error results are shown in Fig 10.

Implementing a mining concrete mixer truck simulation model in MATLAB/Simulink requires the comprehensive use of components such as vehicle dynamics, fluid dynamics, mixing system, control system, and interference model. This is followed by multiple steps such as model design, component selection, parameter configuration, simulation settings, and result analysis. The performance and reliability of the model can be ensured by configuring each component in detail, setting the parameters and simulation environment correctly, and performing system verification and optimization. Documentation and version control further enhance the repeatability and transparency of the model, making it more reliable in practical applications. The repeatability and transparency of the model can be improved by using specific blocks and libraries, optimizing the parameter settings of the model, and enhancing the documentation and version control of the model. As shown in Fig 10A, under the disturbance of fluid sloshing in the tank, the speed tracking control performance of the tank truck using the traditional PID control method is significantly reduced, especially when the tank truck enters the 20˚ ($t$ = 75s) soft mud slope and leaves the 20˚ ($t$ = 275s) soft mud slope, the speed tracking error reaches 0.47km/h, while the speed tracking error of the tank truck using the elastic FA algorithm to optimize the filter adaptive PID control is always kept within 0.3km/h, as shown in Fig 10B. The comparative study shows that the speed tracking accuracy of the concrete tank truck using the proposed control method is consistent with the tracking accuracy without interference, showing that the proposed control method has stronger robustness. This is mainly because the proposed control method can automatically control the PID control parameters through the elastic FA algorithm optimization mechanism, and has a stronger ability to adapt to various working conditions to achieve high-precision speed tracking control driving; on the other hand, the use of the disturbance observer can realize online compensation of system disturbances (the observation results are shown in Fig 9C), further improving the system's anti-disturbance ability.

Fluid sloshing has a significant impact on the stability and safety of mining concrete mixer trucks, which may cause vehicle roll, increased difficulty in handling, reduced concrete quality, and increased accident risks. Improving speed tracking performance can effectively mitigate these risks, ensure stable vehicle driving, reduce fluid disturbances, optimize the concrete mixing process, and enhance overall safety. By improving the control system and implementing real-time adjustment and feedback mechanisms, the accuracy and robustness of speed control can be significantly improved, thereby improving the stability and safety of mining concrete mixer trucks.

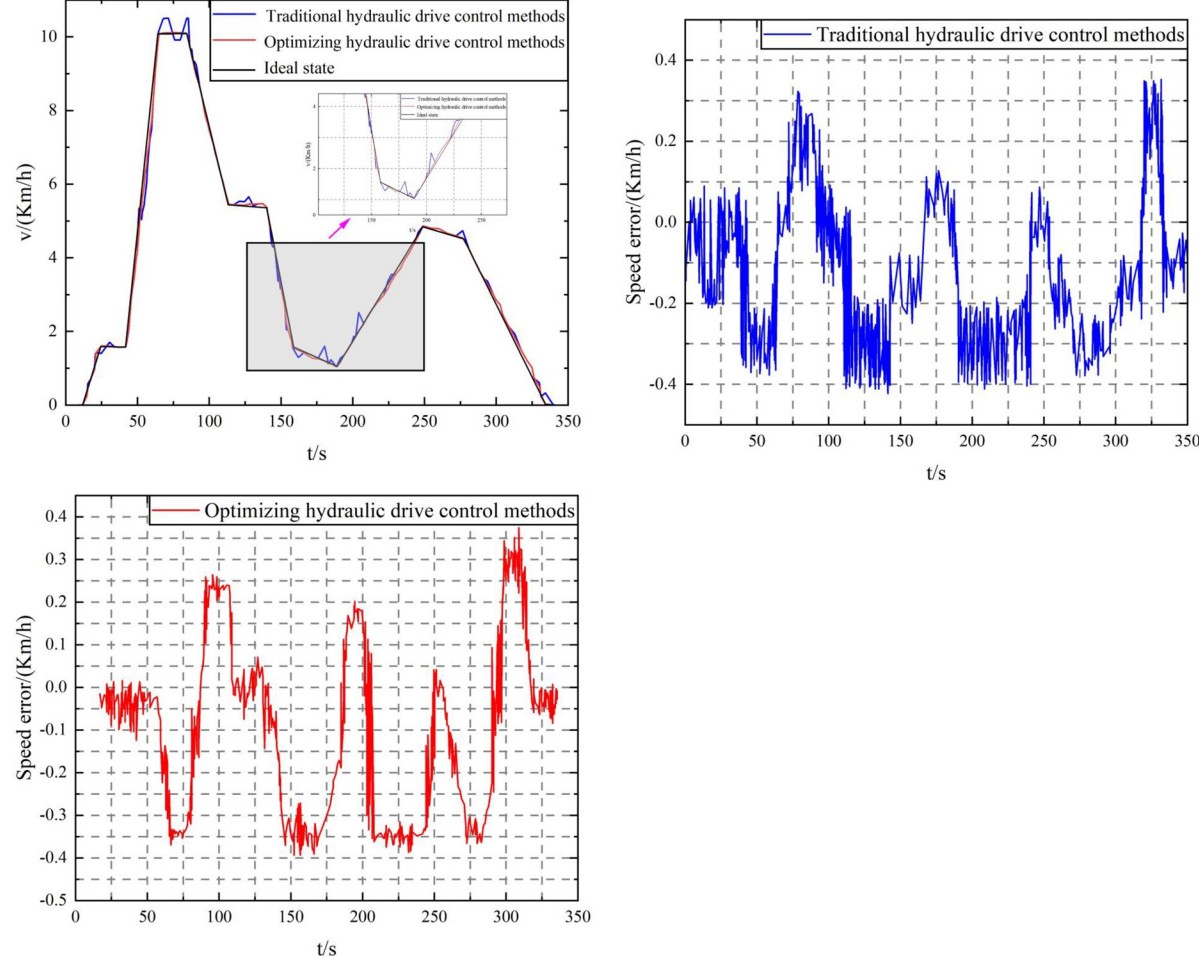

**Fig 10. Continuous driving data of mining concrete mixing tank truck.** (a) Speed tracking, (b) Speed tracking error of traditional hydraulic drive control method, (c) Optimizing the speed tracking error of the hydraulic drive control method.

## 4.3 Simulation model verification

The filling rate $k$ is defined as the ratio of the liquid depth $h$ to the tank radius $R$, that is, $k = h/R$. The liquid sloshing equivalent model takes the second order. To verify the accuracy of the established simulation model of the mining concrete mixing and transport tank truck, the KJCJ-4 mining concrete mixing and transport tank truck was tested. Due to space limitations, only the test of the single lane shifting condition when the filling rate $k = 1.5$ and the vehicle speed $v = 50$km/h are listed here. The comparison of the vehicle operating parameters of the simulation and actual vehicle tests is shown in Fig 11.

Fluid sloshing is usually modeled as the movement of a liquid with a free surface in a container, which may lead to a series of dynamic problems, especially the fluctuation of lateral acceleration, which will affect the stability and safety of the vehicle. For a more accurate simulation, we use CFD software [47] to simulate the movement of liquid in the tank truck, set boundary conditions (such as the geometry of the tank truck, and the physical properties of the liquid), and perform dynamic simulation. Finally, the CFD simulation results are imported into Simulink for joint simulation.

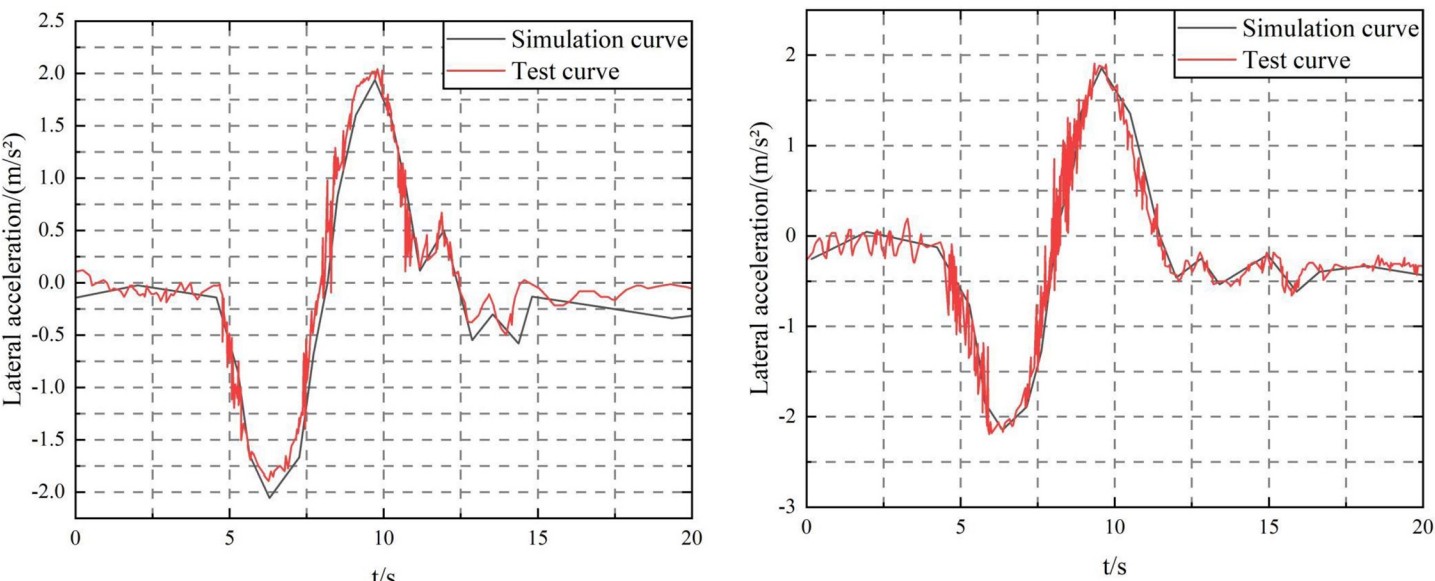

**Fig 11. Comparison of vehicle operating parameters between test and simulation.** (a) Lateral acceleration curve of mining concrete mixing tank truck, (b) Lateral acceleration curve of a mining concrete mixing tank truck under fluid sloshing interference. Note: Single lane shift condition when filling rate $k = 1.5$ and $v = 50$ km/h.

Fluid sloshing has important potential effects on the stability and safety of mining concrete mixer trucks, covering multiple aspects such as lateral acceleration fluctuation, vehicle stability, driver control difficulty, structural fatigue, and material quality uniformity. Accurately capturing these effects in the simulation model is crucial for optimizing vehicle design, developing effective control strategies, improving the effectiveness of driver training and assistance systems, and conducting safety assessments and cost-benefit analyses. Fluid sloshing can cause lateral acceleration fluctuations in the tank truck during driving, especially when turning and making sudden stops and turns. Fluctuating lateral acceleration increases the risk of vehicle loss of control and rollover. Fluid sloshing can also affect the center of gravity and motion characteristics of the vehicle, thereby affecting the stability of the vehicle. The sloshing liquid mass can cause the vehicle to have unstable motion patterns, such as swinging and tilting. It will also increase the difficulty for the driver to control the vehicle, especially in complex terrain or bad weather conditions. The fluctuating lateral force will exert additional torque on the steering wheel, making the vehicle more difficult to control. As shown in Fig 11B, through the above simulation steps, fluid sloshing interference is introduced into the lateral acceleration curve of the mining concrete mixer transport tanker.

It can be calculated that the root mean square error is 0.05 and the relative error is 6.2%. As shown in Fig 11, the actual vehicle test results and the parameter curve obtained by simulation have good consistency in terms of change trend and change range, and the relative error is within 7.2%. Therefore, it can be considered that the established semi-trailer tanker simulation model is correct and reliable, and can be used as the basis for the next step of research.

The elastic FA optimization filter adaptive PID control is compared with the most advanced open-pit mine vehicle control technology (taking model predictive control and sliding mode control as examples). Model predictive control needs to solve an optimization problem at each sampling moment to find the optimal value of the current control input. This optimization process usually involves multiple variables, constraints, and prediction models, so the computational complexity is high. When switching the control state, sliding mode control may cause

the control signal to switch frequently, resulting in high-frequency jitter. In addition, the relative error of model predictive control is 8.3%, and the relative error of sliding mode control is 7.9%, both of which are higher than the values of elastic FA optimized filter adaptive PID control, so it can once again highlight the necessity of research on elastic FA optimized filter adaptive PID control.

## 4.4 Simulation test and analysis

Considering the danger of accurate vehicle testing at high speeds, this project uses simulation testing methods for analysis. The test is divided into two cases: (1) the tank is filled with liquid, and the liquid density is $\rho = 580\text{kg/m}^3$; (2) the cargo carried by the tank is solid, except that it has no fluidity and other physical properties and mass are the same as those in the first case. The tank capacity is 5.5 cubic meters, and other main simulation parameters of the vehicle are shown in Table 1.

**4.4.1 Step steering condition.** The vehicle filling rate $k = 1.5$, the vehicle speed $v = 40\text{km/}$h, the steering starts from 1s, and the steering angle of the front wheel of the tractor reaches 4° at 1.8s. Two liquid and solid conditions tests are carried out, respectively, and the simulation curve is shown in Fig 12.

**Table 1. System parameters.**

| Serial number | Parameter | Index | Unit |
|---|---|---|---|
| 1 | 8800×2000×2450(±50) | Overall dimensions (length × width × height) | mm |
| 2 | 14000 | Total Weight | kg |
| 3 | ±43 | Articulation Angle | ° |
| 4 | Medial:4150 Outer Side:6200 | Turning radius | mm |
| 5 | 320 | Minimum ground clearance | mm |
| 6 | 15/20 | Approach/departure angle | ° |
| 7 | 15 | Maximum climbing angle | ° |
| 8 | 3870×1685 | Wheelbase × Track | mm |
| 9 | 154/2200 | Engine power/speed | kW/rpm |
| 10 | 180 | Diesel tank capacity | L |
| 11 | Articulated, four-wheel drive | Chassis form | / |
| 12 | 23.5 | Maximum driving speed | km/h |
| 13 | 11(25km/h) | Braking distance | m |
| 14 | Service, auxiliary, and parking brakes | Braking form | / |
| 15 | 5.5 | Geometric capacity | m³ |
| 16 | 0~4 | Stirring capacity | m³ |
| 17 | 1~3 | Mixing drum speed when traveling | r/min |
| 18 | 0~18 | Mixing drum discharge speed | r/min |
| 19 | 0~10.5 | Mixing drum tilt angle | ° |
| 20 | ≥0.65 | Discharging speed | m³/min |
| 21 | 920~1800 | Discharge port height | mm |
| 22 | ≥2.7 | Feeding speed | m³/min |
| 23 | 2350 | Feed inlet height | mm |
| 24 | Φ560 | Feed port size | mm |
| 25 | 170 | Effective capacity of hydraulic oil tank | L |

Note: Some data comes from the Internet.

Note: $\frac{dU}{dt}$ represents the lateral acceleration of the semitrailer/(m·s⁻²); $h$ is the liquid depth/m, $\delta$ is the front wheel steering angle of the tractor/rad, $u$ is the vehicle speed/(m·s⁻¹), $\theta$ is the articulation angle/rad, and $\omega_1$ is the yaw rate of the tractor/(rad·s⁻¹).

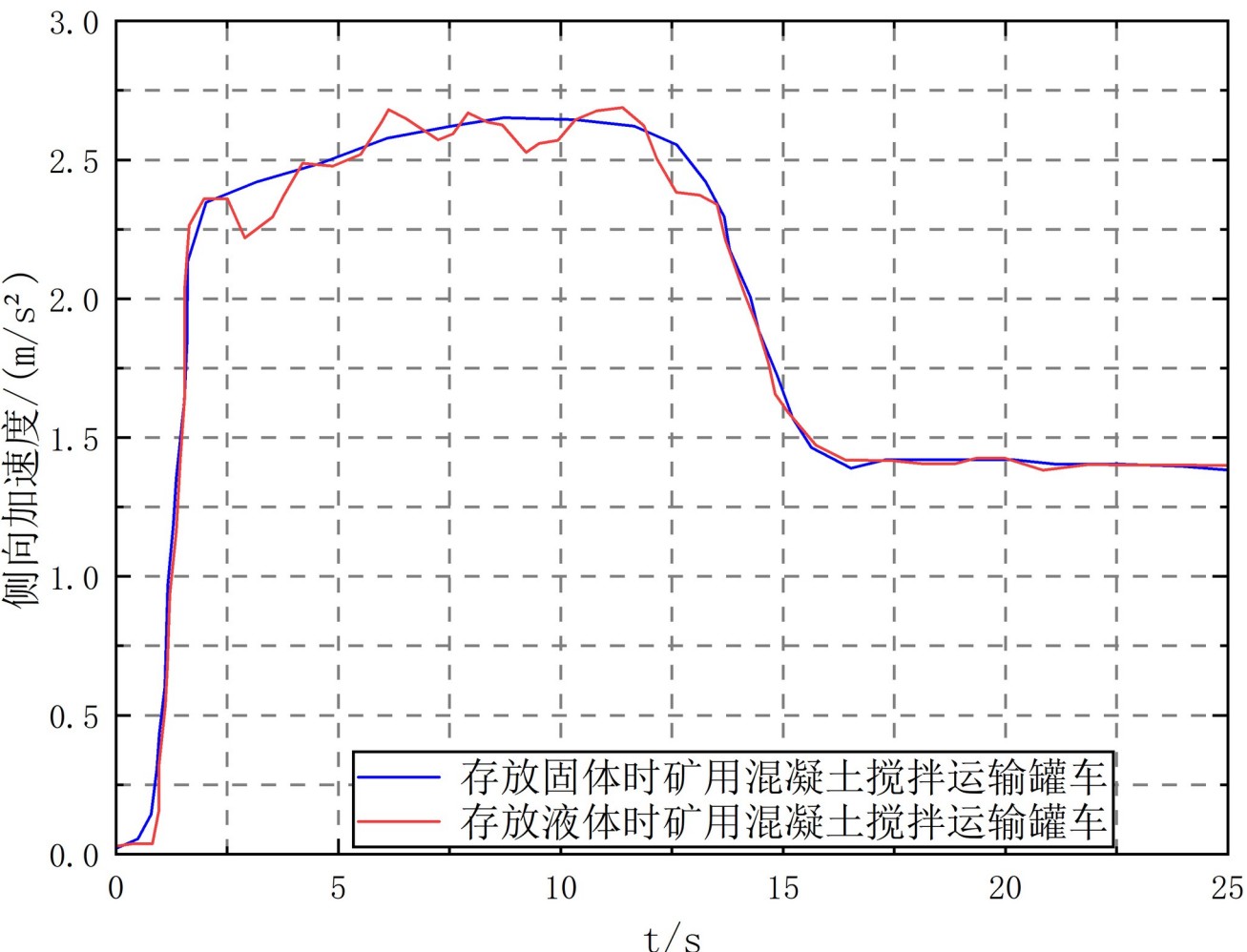

**Fig 12. Lateral acceleration curves of mining concrete mixing tank trucks when loading different objects.** Note: Vehicle speed $v$ = 40 km/h, filling rate $k$ = 1.5.

As can be seen from Fig 12, under the step steering condition, the liquid sloshing has a limited effect on the lateral acceleration and articulation angle of the vehicle but greatly impacts the vehicle roll angle. This is because the liquid flows outward under the action of centrifugal force, which increases the lateral displacement of the liquid center of mass. At the same time, the dynamic pressure generated by the relative movement with the tank body also increases the roll angle. The torque generated by the two on the tank body is compared as shown in Fig 13. As calculated from Fig 13, the positive peak torque of the dynamic pressure acting on the tank body has reached 56.9% of the steady-state value of the lateral displacement of the liquid center of mass. Therefore, the influence of the liquid dynamic pressure on the tank body must be considered when studying the transport tank truck.

The vehicle's rollover is determined by whether the lateral load transfer rate (LTR) [48, 49] is ±1. According to the literature [50], the drive axle of the tanker is the first axle to roll over.

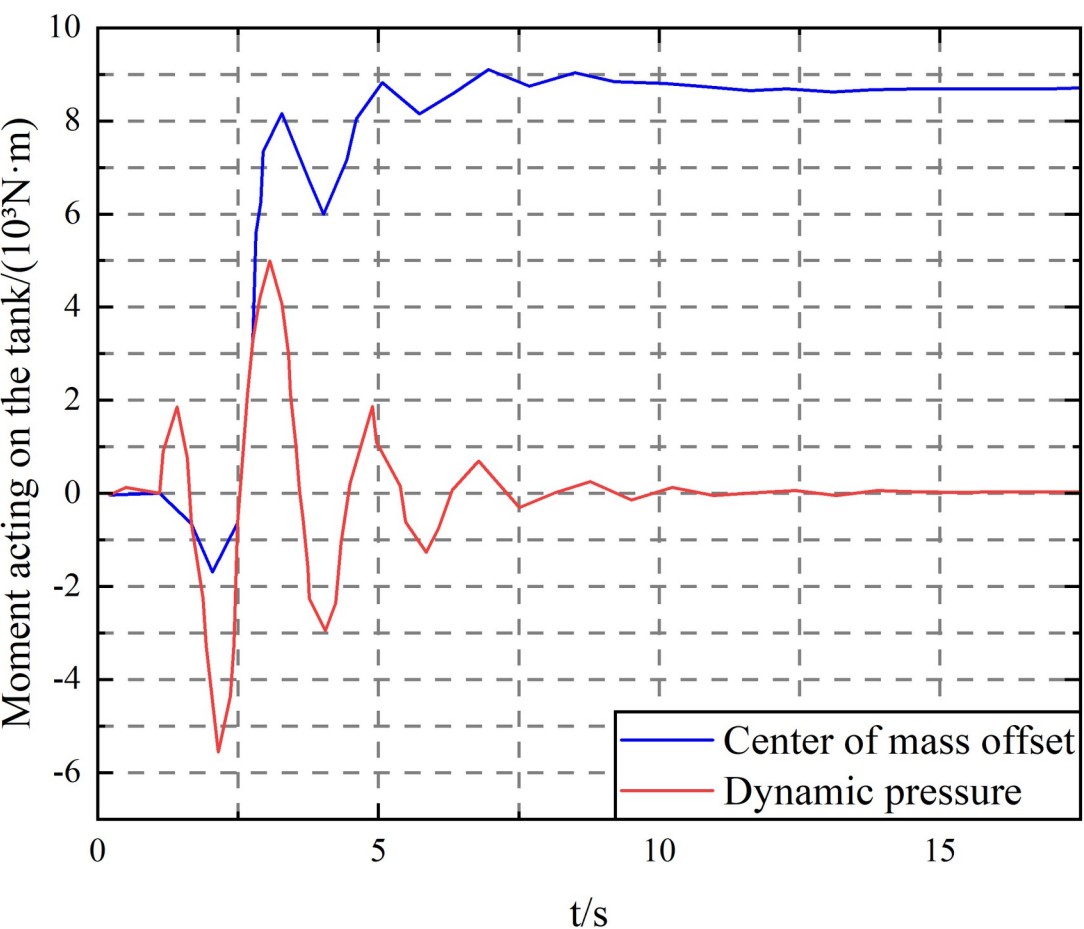

**Fig 13. Liquid center of mass deviation and dynamic pressure moment on the tank.**

The lateral load transfer rate (LTR) of the tractor drive axle is defined as:

$$LTR = -\frac{2(D_r\dot{\varphi} + C_{\varphi r}\varphi)}{M_1 d_2} \tag{51}$$

Where $d_2$ is the wheelbase of the tractor drive shaft, m.

Under the condition that the front wheel steering angle of the transport tanker remains unchanged, the rollover speed thresholds of liquid and solid vehicles with different filling rates are shown in Fig 8.

As shown in Fig 14, under the step-turn condition, the liquid rollover speed threshold is lower than that of solid cargo, indicating that liquid is more likely to roll over, and the difference between the two is the largest near the filling rate $k = 1.5$, indicating that under this filling rate, the impact of liquid sloshing on vehicle driving is the most serious. Therefore, in actual transportation, working conditions near the filling rate $k = 1.5$ should be avoided as much as possible.

**4.4.2 Double lane shifting condition.** Under the working conditions of vehicle speed $v = 60$km/h, filling rate $k = 1.5$, and input of the front wheel steering angle of the tractor, as shown in Fig 9, the vehicle driving parameter curve is shown in Fig 15.

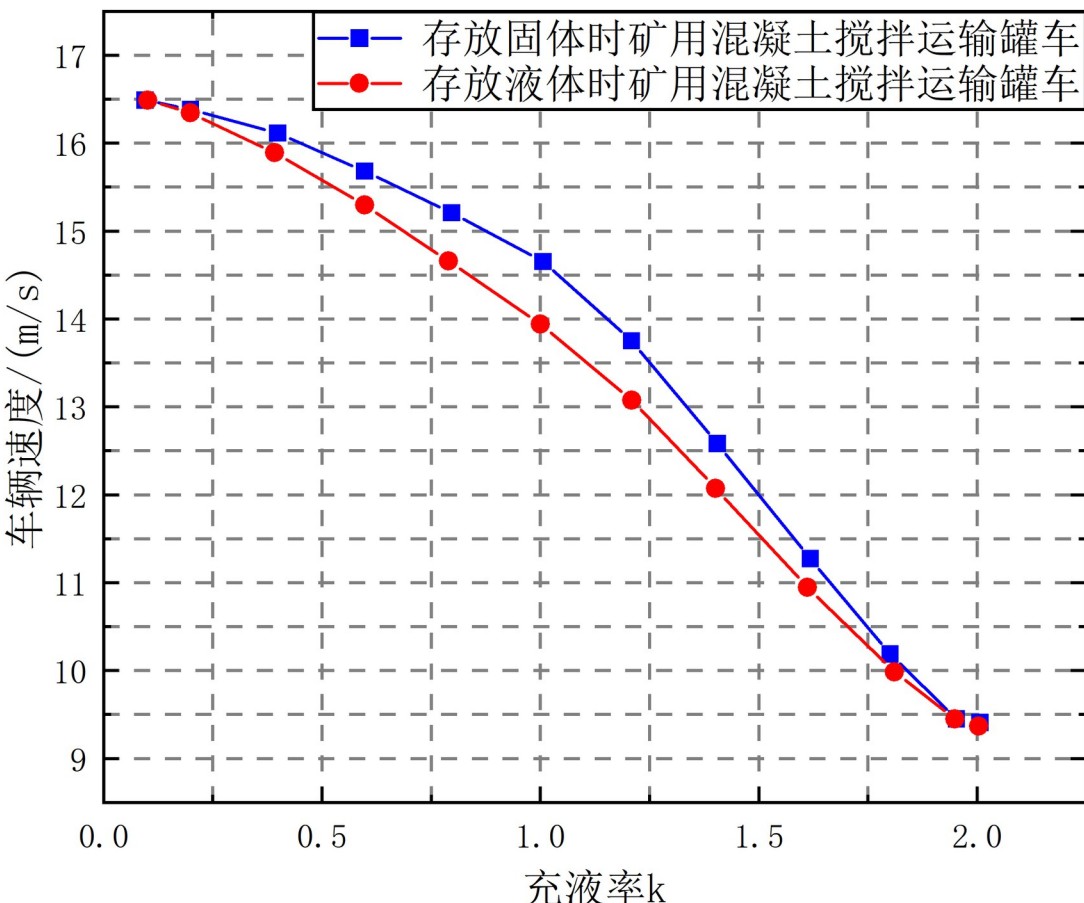

**Fig 14. Rollover speed thresholds at different filling rates.** Note: δ = 4˚ step turn.

As shown in Fig 16, under the double lane change driving condition, liquid sloshing has little effect on the vehicle's roll angle, but it has an impact on the vehicle's lateral acceleration, articulation angle, and articulation angular velocity, increasing its peak value and significantly aggravating the fluctuation. Since the articulation angle and articulation angular velocity can effectively characterize the "folding" and "shimmy" instability of the semitrailer, it means that liquid sloshing reduces the vehicle's driving stability.

The vehicle speed is increased to 65 km/h, and the simulation of liquid filling rate $k = 1.5$ and $k = 2.0$ is performed, respectively. The vehicle's articulation angular velocity and LTR curves are shown in Fig 17.

As can be seen from Fig 17, when the vehicle speed $v = 65$km/h and the filling rate $k = 1.5$, although the LTR is within the range of ±1 and the vehicle does not roll over, its articulation angular velocity has shown equal amplitude oscillation, indicating that the vehicle has experienced "shimmy" instability under this condition. When the filling rate $k = 2.0$, the vehicle can safely complete the double lane change action. This is because when $k = 1.5$, the liquid sways violently, and the sway force and inertia force generated jointly aggravate the vehicle's yaw motion. When $k = 2.0$, the mass of the liquid involved in the swaying is minimal, and the sway force generated has a limited effect on the vehicle's yaw motion, so the liquid and the tank body can complete the double lane change action as the same rigid body.

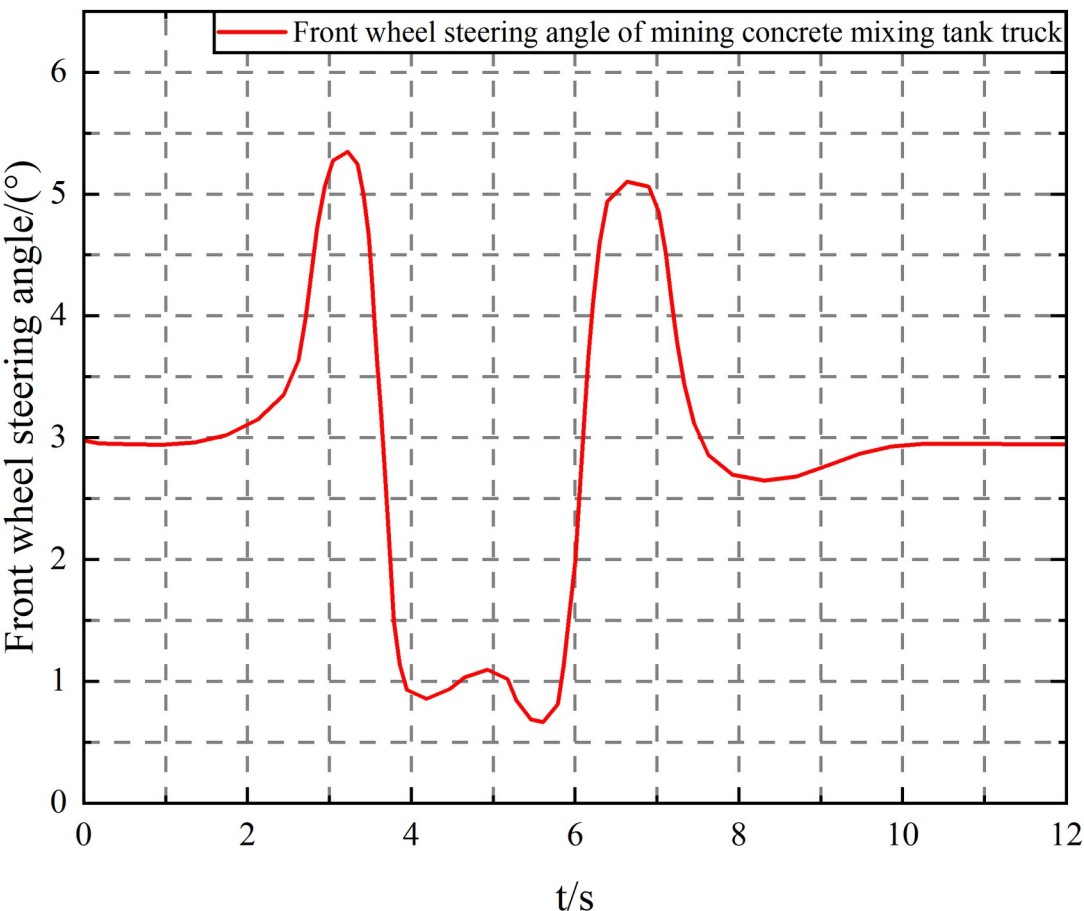

**Fig 15. Front wheel steering angle of the tractor.** Note: Vehicle speed $v$ = 60 km/h, filling rate $k$ = 1.5.

When the vehicle speed increases to 72km/h, the LTR when the filling rate $k$ is 1.5 and 2.0 is shown in Fig 18.

As shown in Fig 18, when $k$ = 1.5 and $k$ = 2.0, LTR> 1 at 17.78s and 20.06s, respectively. This indicates that when the vehicle performs double lane change at high speed, the vehicle will roll over.

## 5. Conclusion and outlook

### 5.1 Conclusion

1. Complex road excitation will cause the fluid sloshing in the tank of a mining concrete mixing tank truck to couple with the dynamics of the vehicle body during driving, thus causing non-axisymmetric problems. By establishing a ground mechanics model and a hydrostatic drive system model for a mining concrete mixing tank truck driving on a complex platform, the model provides detailed physical and dynamic data support for the proposed algorithm. This study fully considers the impact of the amount of subsidence on the stress of the mining concrete mixing tank truck, making the research results more in line with actual operating conditions; to improve the performance of the control algorithm, this paper proposes a filter adaptive PID control algorithm based on the elastic firefly (FA) algorithm after a collective disturbance observer and nominal model filter compensation and uses MATLAB to

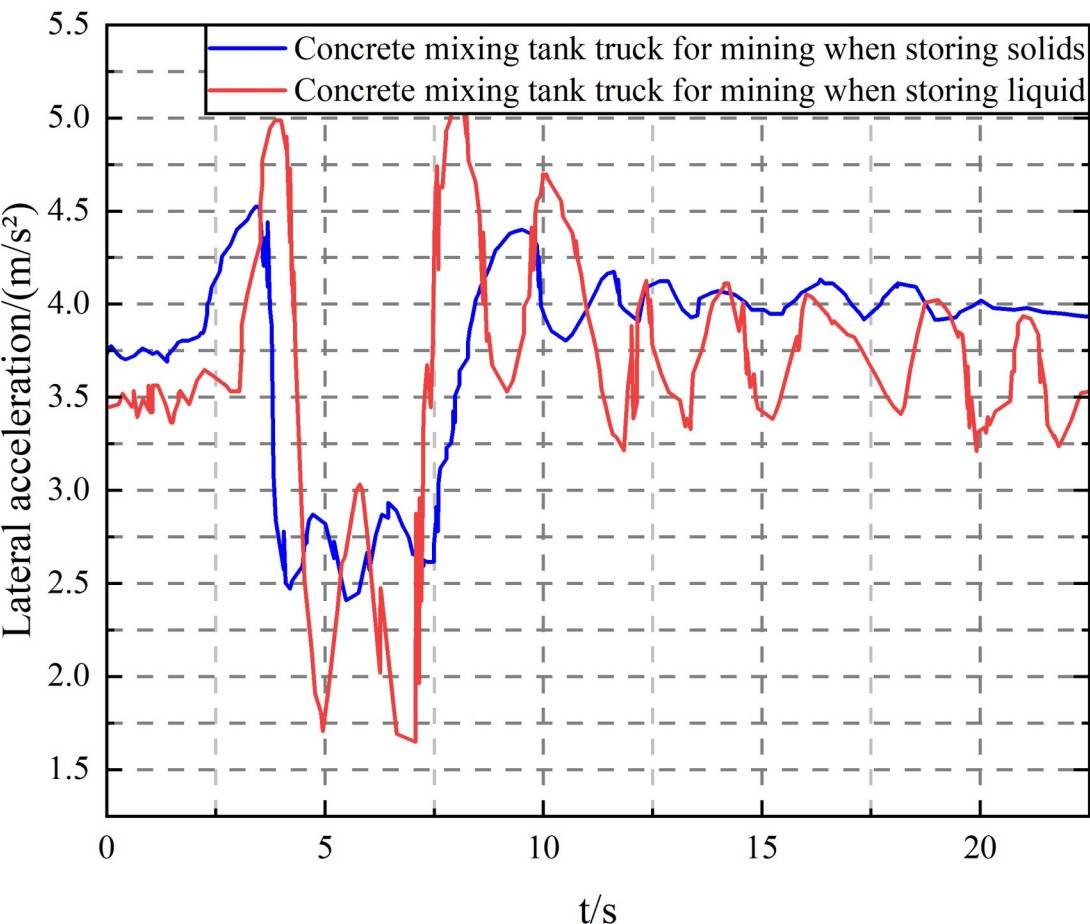

**Fig 16. Vehicle operating parameters for double lane change conditions.** Note: Vehicle speed $v$ = 60 km/h, filling rate $k$ = 1.5.

establish a multi-factor simulation model of a concrete tank truck, and simulates it under step signals, slope platforms and speed changes. The simulation results show that the proposed elastic FA algorithm optimized filtering adaptive PID control algorithm has a good effect on the speed control of concrete tank trucks. The proposed algorithm reduces the stabilization time by 1.6s compared with the traditional PID, and the steady-state accuracy is improved by about 0.03km/h. At the same time, the 20° slope platform acceleration and deceleration robust experiment shows that even under fluid sloshing and white noise interference, the proposed adaptive PID control method can still control the speed error within 0.17km/h, which is 0.2km/h less than the traditional PID, proving its effectiveness.

2. Through the study of liquid sloshing in the tank, a dynamic model of a mining concrete mixing tank truck was established. The simulation results under two working conditions, step turning and double lane shifting, show that the liquid sloshing has different effects on the vehicle driving parameters under the two working conditions. During step turning, the liquid sloshing has a more significant impact on the vehicle's roll angle parameters; during double lane shifting, the liquid sloshing will aggravate the fluctuation of the lateral acceleration but have a more negligible impact on the vehicle's roll angle. When the filling rate $k$ = 1.5 is near, the liquid sloshing is the most violent, which significantly impacts the vehicle's driving stability, so try to avoid driving under this condition.

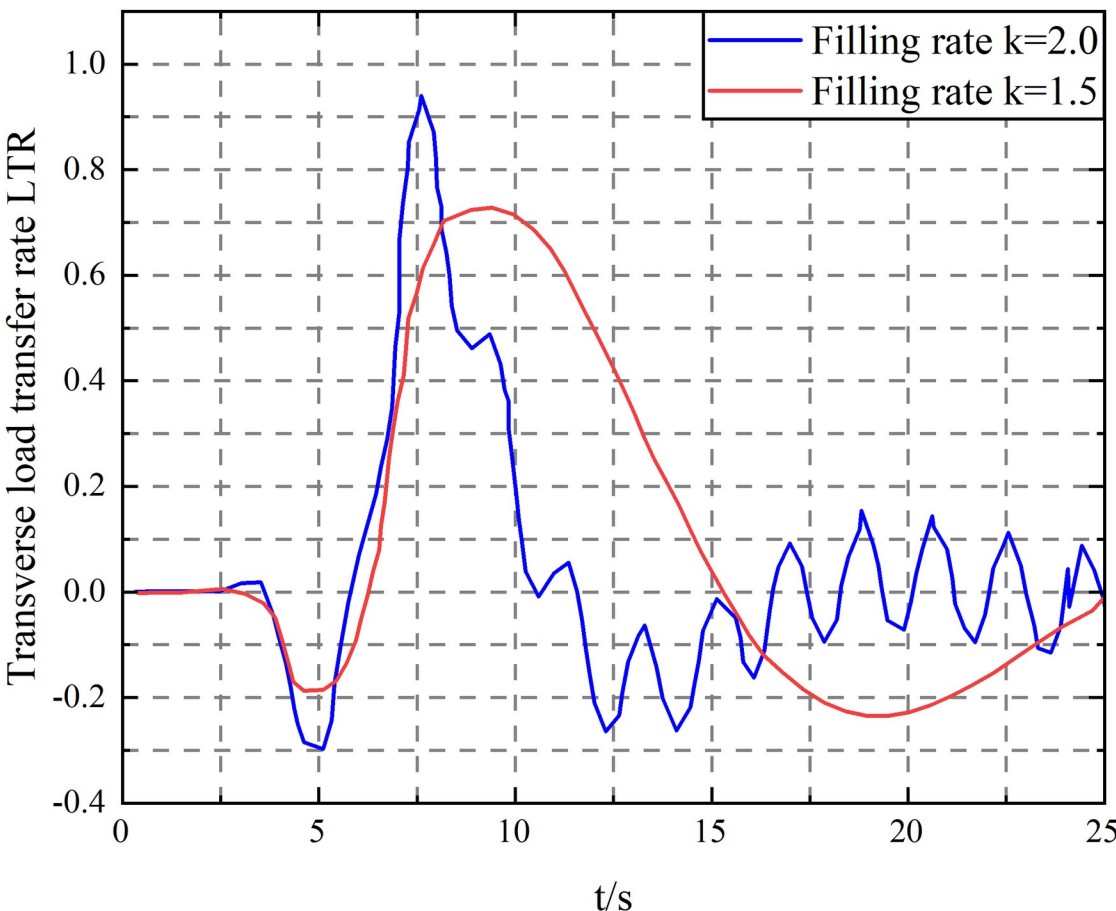

**Fig 17. Transverse load transfer rate curve when vehicle speed _v_ = 65 km/h.**

3. During step turning, the vehicle's instability is manifested as a rollover accident, and the liquid rollover speed threshold is lower than that of solid cargo. In the double lane shifting condition, when the filling rate _k_ = 1.5 is near when the vehicle speed reaches a particular value, it first oscillates and becomes unstable, and then it becomes a rollover accident when the vehicle speed increases; when the total filling rate _k_ = 2.0 is near, the vehicle will directly become a rollover accident.

4. The proposed particle filter PID adaptive control method based on the elastic firefly algorithm can significantly reduce the speed error and improve the steady-state accuracy. Compared with the traditional algorithm, it can effectively avoid the vehicle body coupling phenomenon caused by the influence of complex road excitation, thereby reducing the negative impact of liquid sloshing on vehicle stability and safety, and can greatly improve the efficiency, equipment stability, and safety of open-pit mining operations. Its potential benefits include improving operating efficiency, enhancing equipment stability, improving the working environment, and reducing operating costs. This advanced control method brings a more reliable, efficient, and economical solution to open-pit mining operations. At the same time, by reducing costs, improving safety, protecting the environment, and supporting data-driven decision-making, it further promotes the efficient, safe, and sustainable development of open-pit mining operations.

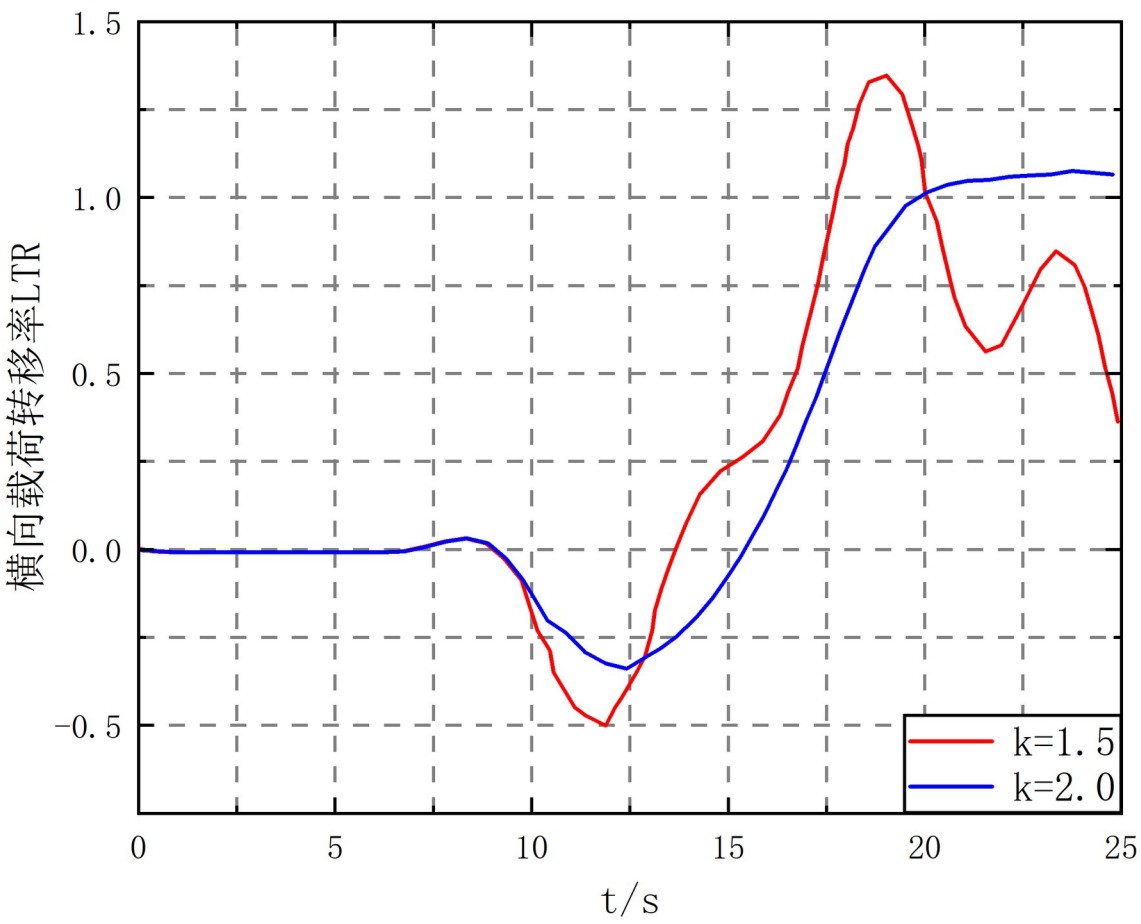

**Fig 18. Transverse load transfer rate curve at vehicle speed of 72 km/h.**

## 5.2 Outlook

1. To verify the effectiveness and performance of the proposed control algorithm under actual working conditions, we plan to conduct experimental tests on a real mining concrete mixer truck. These tests will cover a variety of operating conditions, including different filling rates, vehicle speeds, and road profiles, to evaluate the robustness and adaptability of the control algorithm in real environments. By comparing the experimental results with the simulation results, we will further optimize the control algorithm to ensure its reliability in practical applications.

2. The performance of the control algorithm can be further improved by combining advanced sensing and monitoring technologies. For example, the use of high-precision accelerometers, gyroscopes, liquid level sensors, and force sensors can monitor the dynamic behavior of the vehicle and the sloshing of the liquid in real-time. By integrating with the Internet of Things technology, remote monitoring and data analysis can be achieved, thereby improving the operational safety and efficiency of the vehicle. Future research will explore how to effectively combine these advanced technologies with the proposed control algorithm to build an intelligent and efficient control system.

3. The hydrostatic drive system is an important component of the mining concrete mixer truck, and its long-term durability and reliability are crucial to the performance of the entire vehicle. Under the proposed control scheme, we will study the long-term use of the components of the hydrostatic drive system, including the wear and failure of key components such as hydraulic pumps, hydraulic motors, and control valves. Through long-term experimental tests and data analysis, we will evaluate the impact of the control algorithm on the durability and reliability of the hydrostatic drive system and propose corresponding improvement measures.

4. Although the focus of this paper is on mining concrete mixer transport tankers, the proposed control algorithm has wide applicability. Future research will be extended to other types of heavy vehicles used in open-pit mines, such as mining dump trucks, excavators, and loaders. These heavy vehicles also face complex dynamic behaviors and harsh operating conditions in the operating environment of open-pit mines. Through verification and optimization on different types of heavy vehicles, we will further improve the versatility and robustness of the control algorithm, providing strong support for the intelligent and automated operation of open-pit mines.

## Supporting information

**S1 File. Minimal data statement.**
(DOCX)

**S1 Data.**
(XLSX)

## Author Contributions

**Software:** Wei Liu.

**Validation:** Xuedong Wang.

**Visualization:** Senlin Chai.

**Writing – original draft:** Chonghui Ren.

**Writing – review & editing:** Guangwei Liu.

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
