## [Decision Letter · Decision Letter 0]

10 Jul 2024

PONE-D-24-25486Research on adaptive hydraulic drive optimization control of concrete mixing tank truck for open-pit minePLOS ONE

Dear Dr. REN,

Thank you for submitting your manuscript to PLOS ONE. After careful consideration, we feel that it has merit but does not fully meet PLOS ONE’s publication criteria as it currently stands. Therefore, we invite you to submit a revised version of the manuscript that addresses the points raised during the review process.

We look forward to receiving your revised manuscript.

Kind regards,

Lalit Chandra Saikia, PhD

Academic Editor

PLOS ONE

Journal Requirements:

2. Please note that PLOS ONE has specific guidelines on code sharing for submissions in which author-generated code underpins the findings in the manuscript. In these cases, all author-generated code must be made available without restrictions upon publication of the work. 

Please review our guidelines at https://journals.plos.org/plosone/s/materials-and-software-sharing#loc-sharing-code and ensure that your code is shared in a way that follows best practice and facilitates reproducibility and reuse.

4. We note that your Data Availability Statement is currently as follows: 

"All relevant data are within the manuscript and its Supporting Information files."

**Additional Editor Comments:**

All the comments of reviewers must be addressed and necessary changes must be made as desired in the revised manuscript.

Reviewers' comments:

Reviewer's Responses to Questions

**Comments to the Author**

1. Is the manuscript technically sound, and do the data support the conclusions?

Reviewer #1: Partly

Reviewer #2: Partly

2. Has the statistical analysis been performed appropriately and rigorously? 

Reviewer #1: I Don't Know

Reviewer #2: Yes

3. Have the authors made all data underlying the findings in their manuscript fully available?

Reviewer #1: Yes

Reviewer #2: No

4. Is the manuscript presented in an intelligible fashion and written in standard English?

Reviewer #1: Yes

Reviewer #2: Yes

5. Review Comments to the Author

Reviewer #1: The control is always an important subject, but the results of the modified technique should be essential. In your research you present in the Introduction chapter the analysis of papers published only by Chinese authors. There a lot of international researchers in that domain. The relations are not clear and must be corrected. The modified proposed technique is very similar with the traditional one and the impact is not important. Also I do not understand the expression of filling rate. This is not clear what does it mean. Also why only 2 values for it. Maybe you should find another changing that could bring better results. In my opinion PID control still the best.

Reviewer #2: Dear Authors here is the comments for the manuscript as attached in the file below:

ABSTRACT:

Your review of the abstract provides a thorough analysis of its content, structure, and effectiveness in conveying the key aspects of the research. Let's organize the review into a more structured format to enhance clarity and readability.

Title: Abstract Review

Language and Consistency:

The abstract is well-written and free of any major language errors. The flow of ideas is logical, and the content is consistent with the title and the expected structure of an abstract. However, there are a few minor suggestions for improvement:

1. Clarify the referent of "it" in the sentence "Then, it analyzes its hydraulic drive control characteristics and structural applications" (presumably referring to the proposed method).

2. Rephrase the sentence "Under filling rates of 1.5 and 2.0, two modes of step steering and double lane change are simulated, respectively" to improve clarity, such as "Step steering and double lane change modes are simulated under filling rates of 1.5 and 2.0, respectively."

Recommendations:

1. Clarify the referent of "it" in the sentence discussing the analysis of hydraulic drive control characteristics and structural applications.

2. Rephrase the sentence about the simulation of step steering and double lane change modes under different filling rates to improve clarity.

6. PLOS authors have the option to publish the peer review history of their article (what does this mean?). If published, this will include your full peer review and any attached files.

Reviewer #1: No

Reviewer #2: **Yes: **Hasan Alqaraghuli

---

## [Author Response · Author response to Decision Letter 0]

13 Aug 2024

Dear Editor and Reviewer:

 On behalfof my co-authors, we thank you very much for giving usanopportunity to revise our manuscript, we appreciate editor andreviewers very much for their positive and constructive commentsand suggestions on our manuscript entitled "Research on adaptive hydraulic drive optimization control of concrete mixing tank truck for open-pit mine".(ID:PONE-D-24-25486R1).

 We have studied reviewer' s comments carefully and have maderevision which marked in red in the paper. We have tried our best torevise our manuscript according to the comments. Attached please findthe revised version, which we would like to submit for your kindconsideration.

 Thank you for your questions about the submission information uploaded with the manuscript. Regarding the mismatch between the funding information provided in the "Funding Information" and "Financial Disclosure" sections, the author has matched the two. We included the Role of Funder statement (The funder participated in the review and editing of this article.) was included in the Manuscripth and Cover Letter.

 We would like to express our great appreciation to you and reviewersfor comments on our paper. Looking forward to hearing from you.

Thank you and best regards.

Yours sincerely,

Chonghui REN

Corresponding author:Chonghui REN

E-mail: renchonghui1229@163.com

Address: College of mining, Liaoning Technical University.

---

## [Decision Letter · Decision Letter 1]

28 Aug 2024

Research on adaptive hydraulic drive optimization control of concrete mixing tank truck for open-pit mine

PONE-D-24-25486R1

Dear Dr. REN,

We’re pleased to inform you that your manuscript has been judged scientifically suitable for publication and will be formally accepted for publication once it meets all outstanding technical requirements.

Kind regards,

Zhihong (Arry) Yao, Ph.D.

Academic Editor

PLOS ONE

Additional Editor Comments (optional):

Reviewers' comments:

Reviewer's Responses to Questions

**Comments to the Author**

1. If the authors have adequately addressed your comments raised in a previous round of review and you feel that this manuscript is now acceptable for publication, you may indicate that here to bypass the “Comments to the Author” section, enter your conflict of interest statement in the “Confidential to Editor” section, and submit your "Accept" recommendation.

Reviewer #1: All comments have been addressed

Reviewer #2: All comments have been addressed

2. Is the manuscript technically sound, and do the data support the conclusions?

Reviewer #1: Yes

Reviewer #2: Yes

3. Has the statistical analysis been performed appropriately and rigorously? 

Reviewer #1: I Don't Know

Reviewer #2: Yes

4. Have the authors made all data underlying the findings in their manuscript fully available?

Reviewer #1: Yes

Reviewer #2: No

5. Is the manuscript presented in an intelligible fashion and written in standard English?

Reviewer #1: Yes

Reviewer #2: Yes

6. Review Comments to the Author

Reviewer #1: I appreciate that you used the comments of the reviewers to improve your paper. The impact of the research is considered now important. The therms in the equations have been explained. The originality is now revealed. The simulations are now explained.he originality is now revealed.

Reviewer #2: Dear Authors,

Thank you for your hard work and efforts in presenting this paper. The scientific quality of your work is commendable.

I would like to suggest making all data analysis and simulation files publicly available. This would allow other researchers to build upon your work and potentially help disseminate your ideas more widely in the scientific community.

7. PLOS authors have the option to publish the peer review history of their article (what does this mean?). If published, this will include your full peer review and any attached files.

Reviewer #1: No

Reviewer #2: **Yes: **Hasan Alqaraghuli

---

## [Editor Report · Acceptance letter]

3 Sep 2024

PONE-D-24-25486R1 

PLOS ONE

Dear Dr. REN, 

I'm pleased to inform you that your manuscript has been deemed suitable for publication in PLOS ONE. Congratulations! Your manuscript is now being handed over to our production team.

Kind regards, 

on behalf of

Dr. Zhihong (Arry) Yao 

Academic Editor

PLOS ONE